



# Validation and accommodation of vortex wake codes for wind turbine design load calculations

Koen Boorsma[1], Florian Wenz[2], Koert Lindenburg[3], Mansoor Aman[4], and Menno Kloosterman[3]

[1]ECN part of TNO, Petten, The Netherlands
[2]IAG University of Stuttgart, Stuttgart, Germany
[3]LM Windpower, Heerhugowaard, The Netherlands
[4]DNV-GL, Bristol, United Kingdom

**Correspondence:** K. Boorsma (koen.boorsma@tno.nl)

**Abstract.** The computational effort for wind turbine design loads calculations is more extreme than it is for other applications (e.g. aerospace) which necessitates the use of efficient but low-fidelity models. Traditionally the Blade Element Momentum (BEM) method is used to resolve the rotor aerodynamics loads for this purpose, as this method is fast and robust. With the current trend of increasing rotor size, and consequently large and flexible blades, a need has risen for a more accurate prediction of rotor aerodynamics. Previous work has demonstrated large improvement potential in terms of fatigue load predictions using vortex wake models together with a manageable penalty in computational effort.

The present publication has contributed towards making vortex wake models ready for application to certification load calculations. The observed reduction in flapwise blade root moment fatigue loading using vortex wake models instead of the Blade Element Momentum method from previous publications has been verified using a 'numerical wind tunnel', i.e. Computational Fluid Dynamics (CFD) simulations. A validation effort against a long term field measurement campaign featuring 2.5MW turbines has also confirmed the improved prediction of unsteady load characteristics by vortex wake models against BEM based models in terms of fatigue loading. New light has been shed on the cause for the observed differences and several model improvements have been developed, both to reduce the computational effort of vortex wake simulations and to make BEM models more accurate. Scoping analyses for an entire fatigue load set have revealed the overall fatigue reduction may be up to 5% for the AVATAR 10MW rotor using a vortex wake rather than a BEM based code.

## 1 Introduction

It is expected that from 2025 offshore wind in Germany can operate without subsidy. Also in the Netherlands tenders without subsidy are emerging. To meet this goal large and reliable wind turbines are needed. At that time the size of wind turbines will be in the range of 13-15MW. For a reliable wind turbine design, dependable aerodynamic models are needed. At this moment roughly three categories of aerodynamic models are available:

1. Blade Element Momentum (BEM) models

2. Vortex wake models





3. Computational Fluid Dynamics (CFD) models

In the industry, the Blade Element Momentum method is the workhorse for wind turbine certification load calculations.
Operating in the atmospheric boundary layer, these design load calculations feature a large number of aerodynamic iterations
to define a representative load envelope. This necessitates the use of efficient but low-fidelity models Schepers (2012). The
development of more advanced codes like vortex wake models for wind turbine applications started in early 2000 Belessis
et al. (2001); van Garrel (2003). Vortex wake models give a more accurate description of the rotor wake aerodynamics, but are
more computationally expensive. Nevertheless with increasing computational power, vortex wake models are posing a good
alternative to BEM. As part of the EU-project AVATAR Schepers (2016) a fatigue load comparison round was performed be-
tween various aero-elastic codes using BEM and vortex wake models. Calculations were done featuring the AVATAR 10MW
rotor in turbulent inflow for a variety of time averaged wind speeds. Two partners independently of each other showed a reduc-
tion of roughly 15% of the blade out of plane fatigue equivalent moments when switching from a BEM to a vortex wake type
model for the evaluation of rotor aerodynamics (keeping the other parameters such as the structural dynamic model the same).
Besides that large differences were found in the implementation of the BEM models. More results are given in the dedicated
AVATAR report Boorsma et al. (2016a). Over the last decades several publications have researched the added benefit of vor-
tex wake models over BEM based models Hauptmann et al (2014); Gupta (2006); Boorsma et al. (2016b) and more recently
Perez-Becker et al. (2019). Some of these feature a validation against wind tunnel data, for which the inflow conditions and
turbine are not always representative for design load calculations on a multi MW wind turbine. Others, such as the mentioned
AVATAR report, feature a comparison between BEM and vortex wake models for more representative conditions, but lack a
validation since experimental data for these conditions are not available. Although one would expect a higher fidelity model to
be more accurate, a validation and verification of the outcome is required to confirm the measured load prediction reduction
form vortex wake type codes. Within this mindset the TKI WoZ VortexLoads project Boorsma et al. (2019c) was started in
which ECN.TNO, DNV-GL, LM Windpower and GE have cooperated towards evaluating and accommodating the applica-
tion of vortex wake models to certification load calculations. A comparison against a 'numerical wind tunnel' or dedicated
CFD simulations in turbulent inflow conditions is carried out and described in section 2. A validation against a large field
measurement database, which comprises of long term measurements to reduce the uncertainty in inflow conditions between
measurements and simulations, is given in section 3. Lastly section 4 describes the impact of running a production load set
with a vortex wake rather than a BEM type code.

## 2   Verification against 'numerical wind tunnel'

CFD simulations are carried out to verify the differences in dynamic loading and the resultant fatigue equivalents between
BEM and vortex wake codes. Hereto a 'numerical' wind tunnel was set-up subjecting a rigid (or non-flexible) version of the
AVATAR 10MW wind turbine model, which originated from the AVATAR project Schepers (2016). The various codes used in
the comparison round and their settings are described first. The first focus is at comparisons in uniform, constant inflow and
vertically sheared inflow, after which several cases with turbulent inflow conditions are studied.





## 2.1 Code descriptions

### 2.1.1 FLOWer

A CFD reference solution using the AVATAR 10MW rotor was calculated with the process chain for simulations of wind turbines developed at the Institute of Aerodynamics and Gas Dynamics (IAG, USTUTT) in the last years, i.e. Schulz et al.

(2016). The main part of the chain is the CFD code FLOWer, which is complemented by different pre- and post-processing tools. The CFD code FLOWer was developed by the German Aerospace Center (DLR) within the MEGAFLOW project Kroll et al. (2000) in the late 1990s. It is a compressible code and solves the three dimensional, Navier-Stokes equations in an integral form with several turbulence models. The numerical scheme is based on a finite-volume formulation for block-structured grids. For the spatial discretization, a second order central discretization with artificial damping, Jameson-Schmidt-Turkel (JST)

Jameson et al. (1981) method, and the 5th order weighted essentially non-oscillatory scheme WENO Jiang and Shu (1996) are available. Time integration is accomplished by an explicit multi-stage scheme. Time accurate simulations use the dual time stepping method as implicit scheme. The pseudo time iterations can be accelerated with the same methods as steady computations.

To close the Navier-Stokes equation several RANS and hybrid RANS/LES turbulence models were implemented in FLOWer.

The turbulence model equations are solved separately from the main flow equations using a full implicit time integration method. The ROT module allows body motions in translating/rotating reference frames for unsteady wind turbine simulations. FLOWer is optimized for parallel computing and uses Message-Passing Interface (MPI). A no-slip wall condition was used on the blade surface without any wall function and a far field condition was applied in the cross flow directions. For the current task the Menter SST $k - \omega$ Menter (1994) based IDDES model Shur et al. (2008) was adopted, and no transition model was

considered, i.e. fully turbulent simulations were conducted. A second order dual time stepping method was adopted for the time discretisation and a five-stage Runge-Kutta scheme was used for every inner-iteration. The JST scheme was adopted for the blade meshes, and 5th order WENO scheme was adopted for the background mesh.

Block structured meshes were generated separately for the blade and background, and they were combined without sacrificing the quality of the meshes by using the Chimera overlapping grid technique Chesshire and Henshaw (1990). A blade

mesh convergence test was performed in a previous study Bangga et al. (2017). The blade mesh is a C-type mesh with $[280 \times 128 \times 192]$ grid cells in the chord, wall-normal and span-wise directions. The first wall-off cell size is less than $3 \times 10^{-6}$m, which satisfies the condition $y_1^+ < 1$. The domain size was set to $[3584 \times 1792 \times 1792]\,m^3$ in the stream-wise ($x$) and two cross-flow ($y$, $z$) directions. The rotating axis was aligned with the $x$ axis and located at the origin, which was at a distance of 1536m from the inlet boundary. The total number of cells for simulations with the rotor were $123.5 \times 10^6$.

For the turbulent inflow cases, the wind fields were generated using the Mann turbulence generator from DTU Wind Energy. The generated turbulence field was injected at $x = -400\,\mathrm{m}$ using a momentum source term Troldborg et al. (2014),

$$f_{Ei} = \frac{\rho u_i'}{\Delta x_n}\left(U_n + \frac{1}{2}u_n'\right)$$



where the subscript 'n' indicates the normal component to the turbulence plane. It is noted that the Gaussian convolution, which was used in Troldborg et al. (2014) to avoid numerical oscillation, was not applied because such oscillations were not observed
with the numerical scheme used near the turbulent plane, i.e. 5th order WENO. The time step was set to be approximately 1deg azimuthal variation of the blades per time step. To account for controller initiated changes in rotation speed and pitch, the variations in rotation speed and pitch were recorded during BEM simulations with controller (featuring the same turbulent wind field) and prescribed to the CFD simulation via approximated Fourier series. More detail about the set-up of the CFD simulations can be found in Wenz et al. (2019); Boorsma et al. (2019b).

For a better agreement between lifting line and CFD simulations, the airfoil data for the lifting line simulations was determined from 3D CFD simulations. To vary the angle of attack seen by the blade sections, the inflow wind speed is artificially increased/decreased by maintaining the rotational speed constant at 9.02 rpm. The effective angle of attack seen by the blade sections are then calculated using the reduced axial velocity method, often denoted as the azimuthal averaging technique, according to Hansen et al. (1997). The method takes the averaged velocity upstream and downstream of the rotor plane, and
linearly interpolates the relative velocity at the rotor plane. The resulting polars include the rotational augmentation effects, hence modeling of these should be disabled in the lifting line simulations. Also the Prandtl effect due to the finite number of blades is implicitly included in the CFD simulations. Therefore the polars for the outboard sections (>70%R) are determined from 2D CFD simulations, as this effect cannot be switched off for the vortex wake simulations.

### 2.1.2   Bladed 4.8

The results provided by DNV GL are based on the BEM code of Bladed 4.8. The BEM code in Bladed 4.8 is completely rewritten and replaces the code used in Bladed 4.7 and lower. Recent public validation work is presented in references Collier and Sanz (2016) and Schepers and Boorsma (2014). The model is based on classical BEM theory where the axial and tangential Glauert momentum equations are expressed in dimensional form instead of non-dimensional factors. Further the dynamic submodels (dynamic wake, dynamic stall, skew wake correction) are fully expressed in state-space form allowing combined
direct integration of structural and aerodynamic states. The aerodynamic and structural states are integrated with a 4th order variable step Runge-Kutta integrator. The engineering correction models used in the Bladed 4.8 BEM code are the Øye and Pitt&Peters dynamic wake model (described in Snel and Schepers (1994)), Beddoes-Leishman dynamic stall model in state-space format, Glauert skew wake correction method, Prandtl tip correction and Glauert corrections for highly loaded rotors.

Next to the classical BEM model, Bladed 4.8 and higher features a fully coupled free wake lifting line model. At present this
code is used for internal purposes only and is not yet commercially released. The theory of the lifting line code is described in Kloosterman (2009). Recent work published with the code is found in Schepers and Boorsma (2014) and Harrison et al. (2018). The implementation in Bladed is however fully coupled to the Bladed multibody model and allows for aeroelastic load simulations. For the turbulent inflow test cases, a time step which is approximately equivalent to 1 step per degree of revolution was applied. Special effort has been made to ensure efficient parallelization and vectorization of the code. It is also
possible to distinguish between wake update frequency and aerodynamic time step, which has a great potential for reduction of computational time Boorsma et al. (2019a).



### 2.1.3 Phatas

The computer program Phatas, 'Program for Horizontal Axis wind Turbine Analysis and Simulation' Lindenburg and Schepers (2000), is developed for the time-domain calculation of the dynamic response and the corresponding loads on a Horizontal Axis wind Turbine. The program Phatas is available as tool in the integrated wind turbine design package FOCUS6 foc (2016). The program Phatas has its own 'internal' BEM based aerodynamic model but is also available in a configuration **PhatAero** that uses the aerodynamics from an external module such as ECN Aero Module.

The internal BEM based aerodynamic model features several engineering extensions such as a dynamic inflow model Snel and Schepers (1994), yaw model Schepers and L.J. (1998); Schepers (1999), root and tip loss model Prandtl and Betz (1927) and a turbulent wake state model based on the formulation of Wilson. For the airfoil data, the modeling of dynamic stall behaviour Snel (1997) and rotational effects on lift Snel et al. (1993) are optional based on Snel's models. The internal model features a recent addition to account for the effects of shed vorticity (Blade Shed Vorticity), which calculates a vortex structure of shed vorticities based on the time history of the lift coefficients and the relative velocities of the airfoils Boorsma et al. (2019a).

The aerodynamic properties of the blade were modelled with the Cl-Cd-Cm tables derived from the CFD calculations of the University of Stuttgart. The options for correction of the lift coefficient for the effects or rotation ('3D effects') was not used. There was no model for dynamic stall used. The blades were modelled with 31 elements over the span, where 29 elements have equal length and the 2 elements close to the tip have half that length. The aerodynamic stagnation from the tower was not included. The time increment was set to give a $1°$ increment in rotor azimuth.

The input settings of phatAERO + ECN Aero Module AWSM are for a 6 diameter total wake length of which 2 diameters are for a free-geometry wake length. The calculations were done with the option to skip the 'odd' aero calls, which reduces the number of aerodynamic evaluations and likewise reduce the CPU needed. The simulations featuring the Blade Shed Vorticity model (indicated by Phatas-BSV) incorporate the effect of 20 shed vortices.

### 2.1.4 ECN Aero Module

The ECN Aero Module Boorsma et al. (2011, 2016b) includes two aerodynamic models, the BEM method similar to the implementation in Phatas Lindenburg and Schepers (2000) and a free vortex wake code in the form of AWSM van Garrel (2003). Both models are lifting line codes, i.e. they make use of aerodynamic look-up tables to evaluate airfoil performance. The set-up allows to easily switch between the two aerodynamic models whilst keeping the external input the same, which is a prerequisite for a good comparison between them. Although the package can be coupled to simulation software that solves the structural dynamics of a wind turbine (FOCUS foc (2016), SIMPACK sim (2018)), the stand-alone option is used simulating a rigid turbine with prescribed operational conditions. The BEM model features a local implementation, i.e. solving the momentum equations separately for each blade element rather than once for a full annulus. Several engineering extensions are used such as a dynamic inflow modelSnel and Schepers (1994), yaw model Schepers and L.J. (1998); Schepers (1999),





root and tip loss model Prandtl and Betz (1927) and a turbulent wake state model (replacement of the theoretical momentum
equation with a linear relation between thrust coefficient and axial induction factor above a value of 0.38 for this parameter).

The Snel dynamic stall model Snel (1997) was applied to all simulations (unless stated explicitly otherwise) and rotational
corrections were disabled. For the free vortex wake simulation, the number of wake points was chosen to make sure that the
wake length was developed over at least 3 rotor diameters downstream of the rotor plane. The wake convection was free for the
first 2 wake diameters downstream of the rotorplane. For the remaining wake length, the blade averaged induction at the free
to fixed wake transition is applied to all wake points. For both aerodynamic solvers approximately 20 elements in spanwise
direction were used. The spanwise discretization in AWSM features a cosine distribution, whereas this is linear for BEM
featuring half the spacing at the tip. For the turbulent inflow calculations, the time step was kept at the approximate equivalent
of $1°$ azimuth for both the BEM and AWSM simulations. Wake reduction Boorsma et al. (2018) (reducing the shed vorticity
spacing) was applied after approximately half a diameter convected wake, skipping 9 shed vortices to end up with an effective
distance of $10°$ azimuth between the shed vortices in the remaining part of the wake.

## 2.2 Constant uniform and sheared inflow

Four uniform inflow cases were simulated following part of the power curve, as summarized in Table 1. The resulting load

**Table 1.** Summary of uniform and sheared inflow comparison cases

| Case type | Wind speed $U_\infty$ | Pitch angle | Rot. speed | Shear expon. | Tip speed ratio $\lambda$ | Angle of attack $\alpha^\dagger$@80%R | Axial ind. factor $a^\dagger$@80%R |
|---|---|---|---|---|---|---|---|
| | [m/s] | [°] | [rpm] | [-] | [-] | [°] | [-] |
| uniform | 4.0 | 0.00 | 6.0000 | - | 16.2 | -1.0 | 0.28 |
| uniform | 5.0 | 0.00 | 6.0000 | - | 12.9 | -0.1 | 0.25 |
| uniform | 6.0 | 0.00 | 6.0000 | - | 10.8 | 0.9 | 0.23 |
| uniform | 8.0 | 0.00 | 6.8738 | - | 9.3 | 1.9 | 0.21 |
| shear | 10.5 | 0.00 | 9.0218 | 0.2 | 9.26 | 1.7 | 0.20 |
| shear | 14.0 | 6.06 | 9.6000 | 0.2 | 7.39 | -1.4 | 0.04 |

$^\dagger$ estimate

comparison is given in Figure 1 for the radial distribution of chordnormal force, plus the deduced integral aerodynamic vari-
ables axial force, and power. It is observed that generally speaking a good agreement is found between lifting line and CFD
simulations, which is attributed to the polars being generated from 3D CFD simulations as described in section 2.1.1. The good
agreement is a prerequisite for a successful comparison of unsteady aerodynamics in sheared and turbulent inflow conditions.

In addition to that, also two vertically sheared inflow cases were simulated (see also Table 1). Looking at the flapwise
blade root moment variation in Figure 2a we can observe differences between the predicted amplitudes of the codes. These
differences grow larger for the underlying axial induced velocities in Figure 2b. Here it is noted that 'lifting line variables' such
as angle of attack and induced velocities are not available for the CFD results and hence are not displayed.





(a) Distribution of chordnormal force, $U_\infty$=4 m/s

(b) Distribution of chordnormal force, $U_\infty$=5 m/s

(c) Distribution of chordnormal force, $U_\infty$=6 m/s

(d) Distribution of chordnormal force, $U_\infty$=8 m/s

(e) Axial force coefficient $Cd_{ax}$

(f) Power coefficient Cp

**Figure 1.** Load comparison in uniform inflow conditions



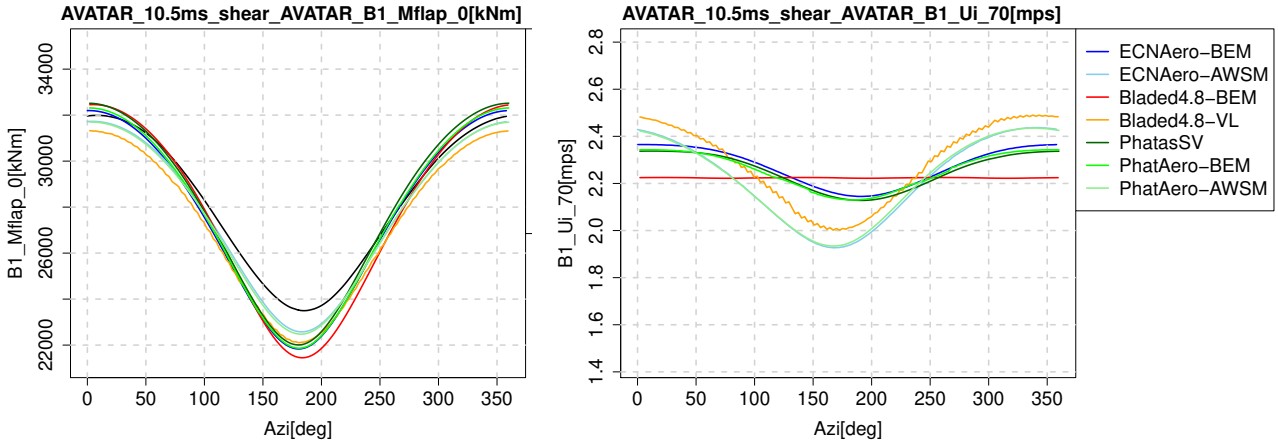

(a) Azimuthal variation of flapwise blade root moment Mflap

(b) Azimuthal variation of axial induced velocity Ui, 70%R

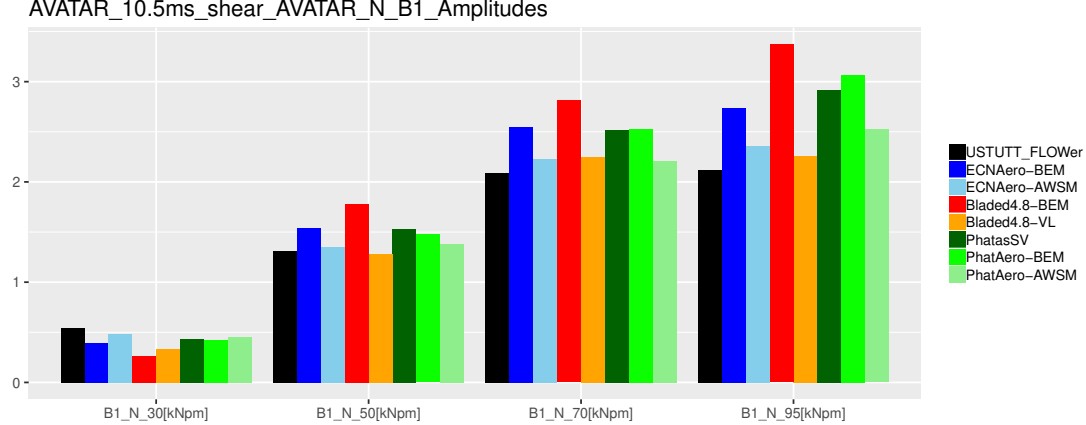

(c) Amplitudes of chordnormal force along the blade span at 30%R, 50%R, 70%R and 95%R

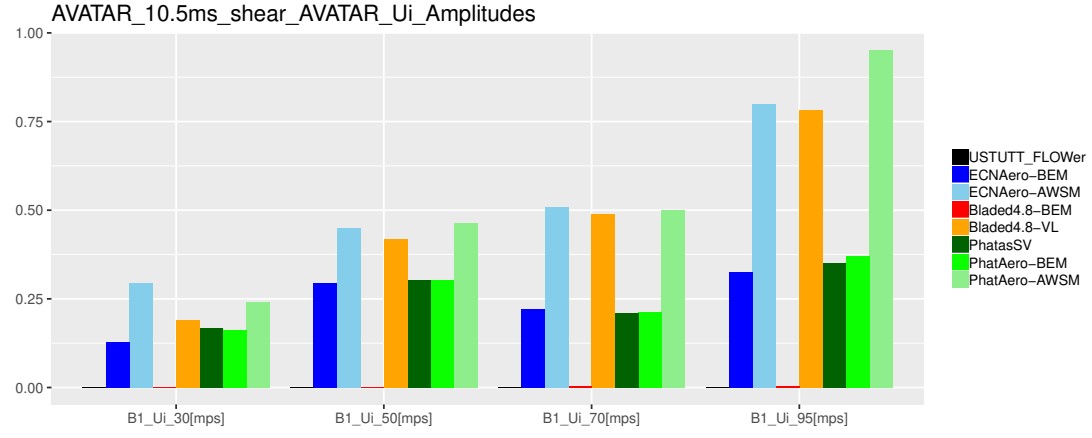

(d) Amplitudes of axial induced velocities along the blade span at 30%R, 50%R, 70%R and 95%R

**Figure 2.** Azimuthal variation and amplitudes in shear, $U_\infty$=10.5 m/s





To better observe the differences the simulation results are post-processed to average values and amplitudes (by evaluating the difference between maximum and minimum values) of the fluctuation along a rotor revolution. The remaining plots of Figure 2 show the results of the amplitude comparison and here we do observe a striking difference between vortex and BEM type codes. The BEM type codes systematically over predict the amplitudes of the normal forces in comparison to the CFD and

vortex wake results (that are in relatively good agreement), consistent along the blade span except for the most inboard station at 30%R. The difference can be traced back to the angle of attack and the underlying axial induced velocity variation in Figure 2d, which can be considered as the 'heart' of lifting line models. For the vortex codes, the axial induced velocity follows more extremely the inflow velocity variations as the blades rotate through the sheared velocity field. Between the BEM codes it can also be observed that whereas some of them predict a substantial azimuthal variation of axial induced velocity, there are also

BEM results where this azimuthal variation along a rotor revolution is almost negligible. It is known that a wide variety of BEM implementations exist, e.g. solving the momentum equations for a whole annulus or per element, not to mention the interaction with a dynamic wake or dynamic inflow model. This example illustrates the effect these implementation differences can have. Application of the dynamic inflow model to the local element induced velocity (as implemented in the Bladed4.8-BEM results following the TUDK model as described in Snel and Schepers (1994)) appears to dampen out induced velocity variations in

non-uniform inflow conditions. The other BEM codes use a similar dynamic inflow model, but the dynamic inflow term is related to the annulus averaged induced velocity rather than its respective element value, which results in better tracking of inflow variations.

In search of a fundamental reason for the difference between BEM and vortex wake codes, calculations were done for various conditions with the Phatas and PhatAero code. These conditions also include 2-bladed and 4-bladed versions of the AVATAR

rotor. For the 2-bladed rotor models the chord distribution is simply 1.5 times larger compared to the chord distribution of the 3-bladed rotor. The 4-bladed rotor model has 75% of the chord distribution compared to the 3-bladed rotor. This 'scaling' gives a similar rotor disk loading except near the blade tip. For all configurations the solidity of the rotor is 0.0408. The result shows that for all operational conditions the 1P variation of the blade root flap moment from the BEM based calculations is larger than from the AWSM calculations. This seems to be related to the axial induction factor, see also Figure 3. Although the

values of the axial induction factor are not distributed homogeneously, a nearly linear trend follows of the ratio between blade root flapwise bending moment variation from BEM simulations compared to the vortex wake (AWSM) simulations. The ratio between root moment variations shows to be quite insensitive to the number of rotor blades or the distance between the vortex sheets of the blades.



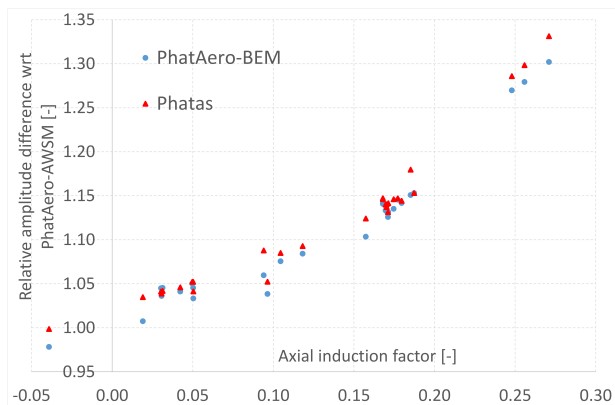

**Figure 3.** Relative difference of Mflap amplitudes from BEM calculations in shear w.r.t. PhatAero-AWSM versus axial induction factor

## 2.3 Turbulent inflow

After exposing the differences between the codes in sheared inflow, the next step is a comparison in turbulent inflow. Six cases were simulated, as summarized in Table 2. To ensure all partners reads the turbulent wind file in the same way and signal

**Table 2.** Turbulent inflow comparison cases

| Nr | Case | Hub wind | Turb. intensity | Length scale | Rot. speed | Pitch angle | Wind seed | Duration |
|----|------|----------|-----------------|--------------|-----------|-------------|-----------|----------|
| [-] |    | [m/s]   | [%]            | [m]          | [rpm]     | [deg]       | [-]       | [s]      |
| 1 | 8ms_fixed | 8 | ~23 | 33.6 | 6.87 | 0.0 | 205 | 400 |
| 2 | 8ms_prscrbd | 8 | ~23 | 33.6 | prscrbd | 0.0 | 205 | 400 |
| 3 | 16ms_prscrbd | 16 | ~17 | 33.6 | prscrbd | prscrbd | 205+offset | 200 |
| 4 | 8msTI10_prscrbd | 8 | ~10 | 33.6 | prscrbd | 0.0 | 205+scale | 400 |
| 5 | 8msCt_prscrbd | 8 | ~23 | 33.6 | prscrbd+1.5 | -1.5 | 205 | 400 |
| 6 | 8msL_prscrbd | 8 | ~23 | 134.4 | prscrbd | 0.0 | 208 | 400 |

processing is in agreement amongst the partners, first an alignment study was performed using a 150 s simulation from the EU AVATAR project Kim et al. (2016), which featured a constant pitch and rotational speed at an average hub height wind speed of 10.5 m/s. The cases summarized in Table 2 are defined in agreement with IEC Class 1A, where wind shear was excluded

from the comparison. The first case featured a constant rotational speed and pitch angle at 8 m/s hub height wind speed. For





the AVATAR turbine the class 1A specification leads to a rather high turbulence intensity of 23% and a length scale of 33.6 m. Seed selection was determined by running six different seeds with BEM and matching the most representative seed to the average values of fatigue, mean and standard deviation over the six seeds. Wind field duration was set to 400 seconds (16 m/s case excepted) based on a compromise between computational expense and a good statistical representation. For the second case, a BEM simulation with the AVATAR controller activated was performed with the wind seed under investigation. The resulting rotational speed and pitch angle variations were recorded and fed to the CFD simulation (this is indicated by the suffix 'prscrbd'). The same procedure was adopted for the other cases. Since the wind speed was below rated, the resulting pitch angle remained constant for this case at $0°$. For the third case the same wind seed was used but the offset was increased to result in an average of 16 m/s hub height wind speed. Acknowledging that a wind seed turbulence box has a constant length, doubling the wind speed effectively means that the simulation duration is halved to 200s. In agreement with IEC Class 1A specifications, the wind speed fluctuations were scaled to match an average turbulence intensity of roughly 17%. For the fourth case, the influence of varying the turbulence intensity was investigated by scaling the amplitude of fluctuations for the same seed to approximately 10%. For the fifth case, the influence of an increased thrust coefficient or axial induction factor was investigated. To this means an offset was applied to the rotational speed and pitch angle variation of the second case. This way the operating angle of attack was not significantly different from the second case and the spanwise variation of the averaged axial induction remained relatively constant. Finally for case six, the influence of a different length scale was investigated by increasing this parameter with a factor of four. The idea behind this case is to mimic the effect of rotor size by changing the turbulence length scale. It is anticipated that the rotational sampling will be different between small and larger rotors, influencing the coherence of the encountered wind gusts. For more info on the case description, please consult the dedicated report from University of Stuttgart Wenz et al. (2019) on this subject.

### 2.3.1 Comparison methodology

In cases with turbulent inflow, besides a statistical evaluation, analysing the development of forces over time is an interesting approach which might give more insight. In order to do this a consistent input of background turbulence in the different codes has to be ensured. In CFD, turbulence is altered as it propagates through the domain until it reaches the rotor, while in BEM and vortex wake models, the flow field, i.e. turbulence, is applied directly to the rotor. Moreover, the propagation in CFD is slowed down in front of the turbine due to the rotor blockage. To allow a time-dependent load comparison between the codes these CFD effects need to be compensated in the lifting line code input. This was achieved by extracting the turbulent velocity field from the empty box CFD simulations and applying a time shift to compensate for the blockage effect. A detailed explanation of the method is given elsewhere Wenz et al. (2020, 2019)

The resulting alignment between the codes was verified by comparing the values of the encountered wind by the blades using virtual 'wind probes' at several radial stations. Generally speaking a good agreement of the encountered wind variation as a function of time was found using this method, indicating that the turbulence structures from empty box and rotor CFD are highly alike, providing similar fluctuations due to the rotational sampling. However it should be realized that although this method comes close, definition of identical inflow conditions between CFD and lifting line codes is impossible and the current





approach is an approximation based on an engineering method. As such small inflow differences between lifting line codes and
      CFD remain.

### 2.3.2   Differences between the models

      To study the differences between the codes the statistics (minimum / maximum / average / standard deviation) over the full
      time serie were determined as well as the 1Hz equivalent loading (based on the rainflow counting procedure using a slope of
m=11) for a large number of variables. Despite the small differences in inflow definition, the 8 m/s fixed case allows to draw
      some interesting conclusions with respect to the effects of modeling differences. A summary of key results from the 8 m/s
      (fixed rotational speed and pitch angle) case are shown in Figure 4. In agreement with results from the AVATAR project and the
      sheared inflow comparison, the induced velocity variation of vortex wake models follows more directly the underlying inflow
      variations than the BEM results. The vortex wake model from Bladed features the same behavior as the previously studied
vortex wake models. As a result, the fluctuations in angle of attack and consequently aerodynamic loads are smaller. This is
      clearly affecting the equivalent sectional load levels at all radial stations (inboard at 30%R often excepted) and hence also the
      blade root moments. The comparison to CFD indicates that generally speaking the vortex codes agree better with CFD than
      BEM judging by the magnitude of load fluctuations and resulting equivalent load levels.

      From previous work Boorsma et al. (2016a) it was hypothesized that part of the observed difference between BEM and
vortex type codes can be explained by the shed vorticity modeling which is implicitly included for vortex wake models but
      not in BEM. A dedicated model to simulate the effect of shed vorticity changes has been developed for the Phatas code, called
      Phatas-BSV, see also section 2.1.3. It is also noted that the indical method from Beddoes & Leishman Leishman and Beddoes
      (1986, 1989) for modeling unsteady sectional aerodynamics includes a part dedicated to modeling shed vorticity effects based
      on Theodorsen's theory Theodorsen (1935). As alternative to Snel's dynamic stall model, this submodel was applied in the
ECN Aero Module (ECNAero-BEM-BL). The resulting fatigue equivalent blade root moments displayed in Figure 5 indeed
      confirm that modeling shed vorticity partly reduces the discrepancy between BEM and vortex wake codes. In addition to that
      it is observed that both the blade shed vorticity model in Phatas (which acts on the induced velocities by modelling a shed
      vorticity structure) and the Theodorsen part of the Beddoes Leishman (which acts on the airfoil coefficients rather than induced
      velocities) result in a similar effect on the fatigue equivalent moments.

Studying the axial induced velocity variations in Figure 4a reveals not only large differences between BEM and vortex wake
      type codes, but also between the different BEM codes. Where the red line shows a nearly constant level, the other BEM codes
      feature more variation with inflow velocity. This was also observed in the sheared inflow comparison as described in section
      2.2, where an explanation for the differences between the BEM codes was given.





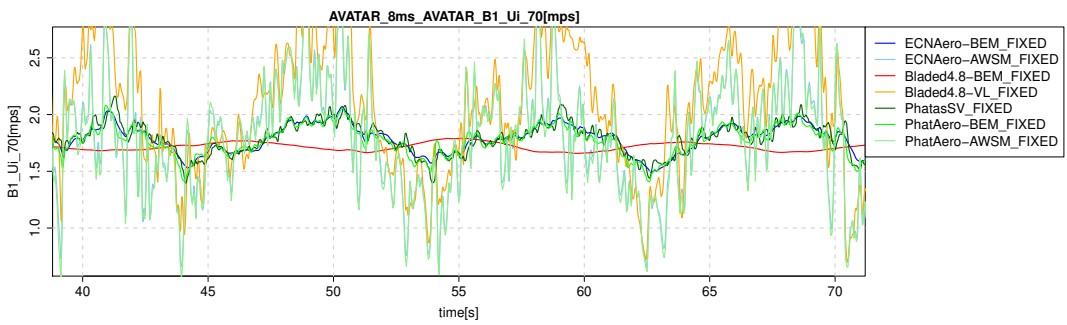

(a) Time trace of axial induced velocity at 70%R

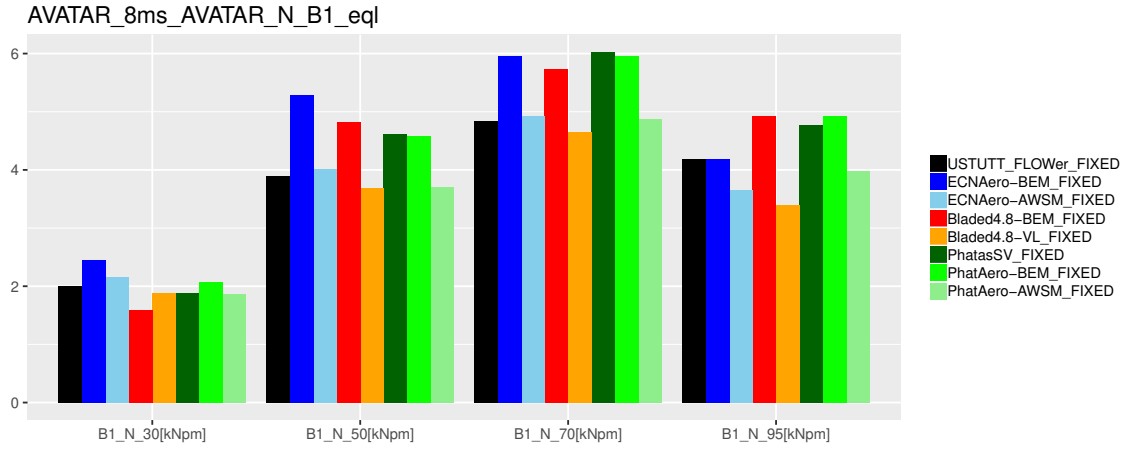

(b) Fatigue equivalent of chordnormal force at 30%R, 50%R, 70%R and 95%R

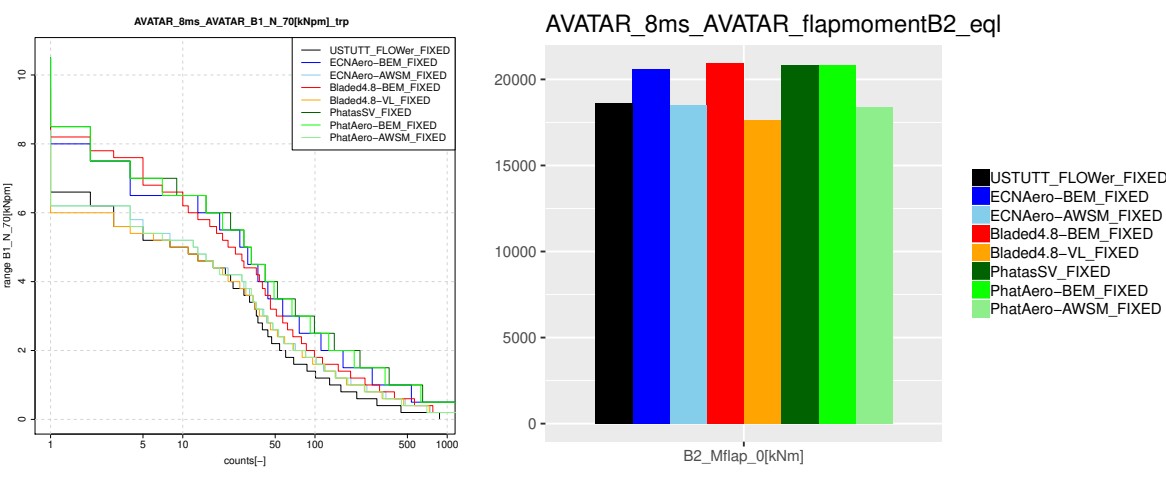

(c) Staircase plot of chordnormal force at 70%R

(d) Fatigue equivalent flapwise blade root moment

**Figure 4.** Key results for the $U_\infty$=8 m/s fixed rotational speed and pitch angle case

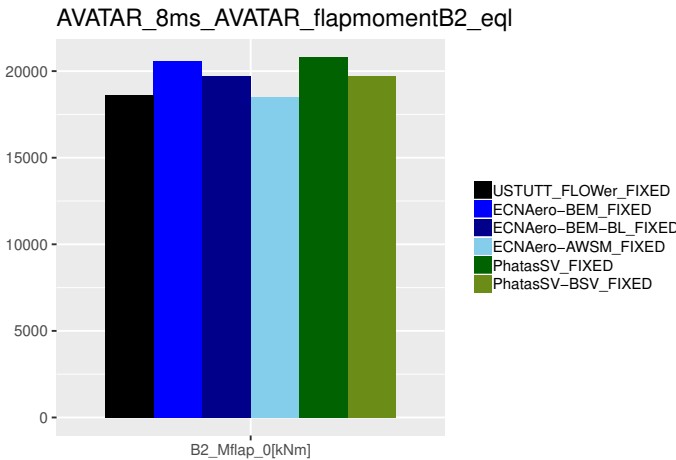

**Figure 5.** Effect of shed vorticity modeling on fatigue equivalent flapwise blade root moment ($U_\infty$=8 m/s fixed)

### 2.3.3 Effect of load case variations

For several cases the earlier mentioned small differences in inflow conditions between CFD and lifting line codes influence the equivalent load levels, as the result of the rainflow counting is dominated by the largest fluctuation over the time series. Therefore it is decided to study the staircase plots from the rainflow counting procedure, which are an intermediate result showing the range of fluctuations versus the number of occurrences or counts. Instead of focusing on the equivalent load level determined by the largest ranges with very few occurrences, statistically it makes more sense to study the ranges with 280 a large number of counts when comparing CFD to lifting line simulations. To compare the results between the codes over the simulated load cases, the staircase plots of the flapwise blade root moments (e.g. Figure 4c) were integrated (starting at a threshold of 10 counts, keeping the logarithmic distribution for the number of counts). A summary of the results is given in Table 3. In agreement with the results of the 8m/s fixed case, the vortex wake results tend to agree well with CFD also for the other cases. Drawing general conclusion on variations between the load cases is complicated because observed differences 285 between the cases can potentially be caused by the difference in specific turbulence boxes (seeds) and the way the rotor blades slice through them. It seems that similar to the shear case, a higher thrust coefficient value results in larger differences between BEM on the one hand and vortex / CFD models on the other hand. The 16 m/s result features a very low thrust (axial induction factor around 0.06), which makes the BEM with shed vorticity modeling come very close to the vortex model, although a rather high unexplained difference remains with CFD. Simulating a higher length scale (mimicking a 4 times smaller turbine) 290 unexpectedly seems to have hardly any impact on the magnitude of the differences between the models.

Although comparison of equivalent loads between CFD and lifting line codes is hindered by small differences in inflow conditions, a comparison between lifting line codes (BEM and vortex) in terms of fatigue loading is deemed useful in the last two columns of Table 3. These numbers also confirm that the shed vorticity modeling in BEM for this 16 m/s case make these





**Table 3.** Relative difference of staircase plot integrated (INT) and fatigue equivalent (EQL) flapwise blade root moments[†]

| Case | INT relative to CFD | | | EQL relative to AWSM | |
|---|---|---|---|---|---|
| | ECNAero-BEM | ECNAero-BEM-BL | ECNAero-AWSM | ECNAero-BEM | ECNAero-BEM-BL |
| | [%] | [%] | [%] | [%] | [%] |
| 8ms_fixed | 22.2 | 10.0 | 1.4 | 13.8 | 8.9 |
| 8ms_prscrbd | 29.6 | 11.2 | 5.6 | 12.5 | 5.4 |
| 16ms_prscrbd | 31.6 | 9.2 | 12.6 | 8.5 | -1.6 |
| 8msTI10_prscrbd | 26.1 | 8.5 | 2.8 | 14.0 | 8.0 |
| 8msCt_prscrbd | 34.8 | 13.4 | 0.8 | 19.4 | 11.0 |
| 8msL_prscrbd | 30.1 | 11.4 | 6.4 | 14.9 | 7.5 |

[†] Averaged over 3 blades, staircase plots integrated from a threshold value of 10 counts.

results come very close to the vortex wake results. And this difference to be at maximum for the high thrust case. It can also
be observed that, although the absolute level of the flapwise fatigue load will decrease with a lower turbulence intensity, the
relative difference between BEM and vortex code type results remains similar. For more details the full report Boorsma et al.
(2019b) about the comparison between lifting line and CFD simulations can be consulted.

### 2.4 Improved induction tracking

From section 2.2 it appeared that a structural difference exists between the predicted load fluctuation amplitudes in vertical
shear from vortex wake (AWSM) and BEM type codes, which correlates with the axial induction factor. Application of an
engineering extension to BEM, accounting for the effect of shed vorticity variation did not yield an explanation for this difference. Most likely the gradual inflow variations in vertical shear are not abrupt enough for the shed vorticity variation to play
an important role. In turbulent inflow (see also Figure 5), shed vorticity can explain part of the observed differences between
BEM and vortex wake modeling. In an attempt to further study the cause for the remaining difference, results from BEM and
vortex wake sheared inflow simulations on the AVATAR rotor have been post-processed to verify compliance with the axial
momentum equations. The one dimensional axial momentum equations constitute a relation between the thrust coefficient Ct
and the axial induction factor a at the rotor disc in the form of

$$Ct = 4a(1-a) \quad \text{and} \quad a = Ui/U \quad , \tag{1}$$

with

310



| Ct | [-] | thrust coefficient |
|----|-----|--------------------|
| a | [-] | axial induction factor at the rotor disk |
| Ui | [m/s] | axial induced velocity |
| U | [m/s] | wind speed. |

To be able to focus on the effect of shear a relatively large shear exponent of 0.75 at 8 m/s hub height wind speed was employed for the investigation. The results in Figure 6a indicate that the BEM simulation complies with the underlying mo-

315   mentum equation as it is supposed to. However the vortex wake results in Figure 6b clearly deviate from this line depending on the azimuthal position, where especially for the outboard stations high thrust coefficients are obtained in combination with a relatively low axial induction factor (lower as would be the case for the theoretical momentum line) for a downward pointing blade featuring the lowest local inflow velocity. It can be shown that for rotors operating at higher induction, BEM theory

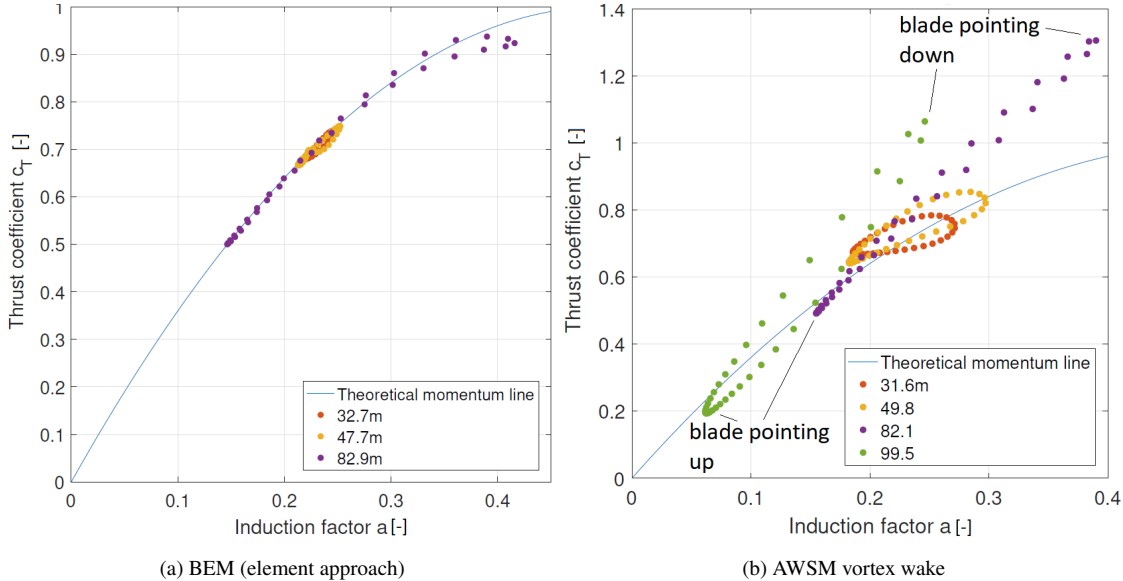

(a) BEM (element approach)                 (b) AWSM vortex wake

**Figure 6.** Comparison of post-processed AVATAR rotor simulation results in heavy shear (U=8 m/s, $\alpha = 0.75$) for several radial stations against the theoretical momentum line

can even predict an increase rather than decrease in axial induced velocity when the blade is pointing downward (6 o' clock

320   position), due to the fact that for the relative high local thrust coefficient the corresponding axial induction factor increase is larger than the local wind inflow decrease (a=Ui/U). This is unlikely to be the case in reality (non-physical), which is backed up by the fact that corresponding vortex wake calculations predict the opposite trend, namely the axial induced velocities to decrease for a lower local inflow speed at the downward pointing blade position.



Acknowledging the fact that the vortex wake model does not obey the momentum equations as implemented in BEM theory,
one may reflect on which shortcoming of BEM theory is responsible for this difference. Several assumptions are made in the
derivation of BEM theory and an inventory was made which specific violations of this theory occur in sheared inflow.

– Radial independence

The influence of neighbouring elements and of the other blades are not taken into account and each annulus is treated
separately. For a spanwise uniform circulation distribution it is acknowledged that intermediate trailed vorticity effects
are absent and as long as the loading differences between the blades are not significant this effect can be neglected as well.
However in sheared inflow, even for a blade that is designed with uniform spanwise circulation distribution, a varying
spanwise circulation distribution will result in trailed vorticity which violates the radial independence assumption.

– Axi-symmetric or uniform inflow conditions

It is well known that BEM theory assumes steady inflow conditions, which relates to the variation of wind velocity along
the longitudinal direction of the streamtube. In addition to this the derivation of the underlying one dimensional axial
momentum equation assumes that the inflow conditions are axi-symmetric (or uniform) with respect to the streamtube
considered. In sheared inflow conditions (or any other non-uniform inflow condition such as turbulent or waked inflow) it
may be clear that this is not the case. It was shown previously that the Betz limit can be exceeded in non-uniform inflow
conditions Chamorro and Arndt (2013). The implication of the violation of axi-symmetric or uniform inflow conditions
may differ between a BEM approach that solves the momentum equations 'annulus averaged' (i.e. using one equation
resulting in the same induced velocity for all the blades) or the more modern local approach that solves the momentum
equation separately for each blade. In the latter case one may ask the question what the azimuthal and radial extent of
the streamtube is, that balances the force exerted by a blade.

Further consideration of the second violation aspect triggered the idea to distinguish between the wind velocity used in BEM
for the purpose of evaluation of the sectional force (blade element part) and the determination of induced velocity (momentum
equations). See also the below displayed axial momentum equation 2, assuming that the tangential induction can be ignored.

$$2\mathrm{a}(1-\mathrm{a})\rho \mathrm{U}^2 2\pi r dr = \sum_B c0.5\rho W^2 c_l(\alpha)\cos(\phi)dr \quad,$$ (2)

where

$$\phi = atan2(W(1-\mathrm{a}),\Omega r), \quad \alpha = \phi - \epsilon \quad \text{and} \quad W = \sqrt{\mathrm{U}^2(1-\mathrm{a})^2 + (\Omega r)^2}$$

with





| | | |
|---|---|---|
| $r$ | [m] | radius of element considered |
| $c$ | [m] | local blade chord at radius $r$ |
| $\rho$ | [kg/m$^3$] | air density |
| $W$ | [m/s] | effective velocity at element |
| $c_l$ | [-] | lift coefficient |
| $\alpha$ | [°] | angle of attack |
| $\phi$ | [°] | inflow angle wrt rotor plane |
| $\Omega$ | [rad/s] | rotor speed |
| $\epsilon$ | [°] | twist plus pitch angle and torsion deformation |

In this equation the blade element part is on the right hand side, in which the wind velocity U is included through the ef-
fective velocity term $W$, the inflow angle $\phi$ and angle of attack $\alpha$. The momentum part is on the left hand side of equation
2. It is noted that in the so called 'annular average' BEM, the element forces are summed over the blades ($\sum_B$) and the cor-
responding annulus has a $360°$ extent, resulting in a single axial induction factor for all blades. The current 'element 'BEM
implementation which is used here solves equation 2 for each blade separately (adjusting the annular volume correspondingly),
resulting in different induced velocities for each blade element in an annulus.

Revisiting the idea to distinguish between the wind velocity used in BEM for the purpose of evaluation of the sectional force
(blade element part) and the determination of induced velocity (momentum equations), it is clear that the local wind velocity
acting at an element quarter or three-quarter chord point should be used for the first part. For the second part it could be argued
to use a wind speed that is representative for the streamtube considered instead of a local point at the element center (as it is
currently implemented). The question is how to define this streamtube and how to define a representative wind speed for it.
Where a CFD or vortex wake simulation considers all spatial wind speed variations by means of a mesh, the momentum theory
in BEM allows for only one. Effectively this is an inherent shortcoming of BEM and it could be argued we have arrived at a
limitation that cannot be overcome. In a first attempt a streamtube is defined that considers an annular sector with azimuthal
extent of $360°$ divided by the number of blades, symmetrically distributed around the element of consideration. A simple five
point average is taken of the wind speed, the five points equally distributed in azimuthal direction at a spacing of $20°$ for the
current example with three blades. Application of this idea to the above highlighted vortex wake simulation yields Figure 7,
which shows an improvement in terms of agreement with the theoretical momentum line (i.e. the result lie closer to this line).

    Implementing the outlined approach in a BEM code has allowed for some further testing in sheared and turbulent inflow as
illustrated in Figure 8. In sheared inflow it is shown that induced velocity amplitudes and consequently the normal forces are
more in line with the vortex wake modeling. The time trace in turbulent inflow (Figure 8c) clearly illustrates the improved track-
ing of induced velocity of the sector wind approach, again very close to the vortex wake result except for the higher frequencies.
The resulting integrated staircase plots (which are the result of the rainflow counting procedure for obtaining fatigue equivalent
loads) show that application of a shed vorticity model (by means of the Beddoes Leishman model ECNAero-BEM-BL instead



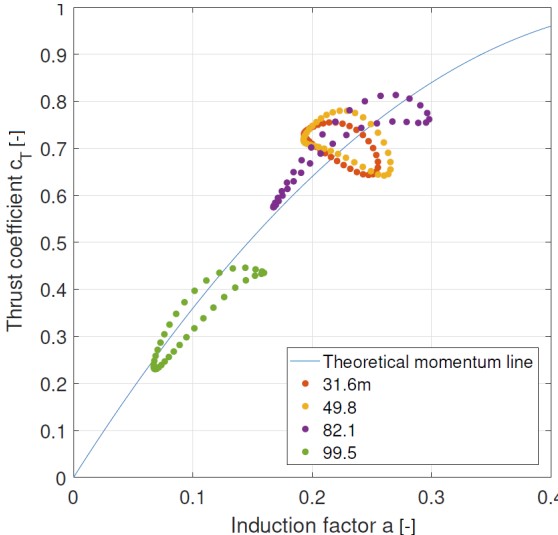

**Figure 7.** Comparison of post-processed AVATAR rotor simulation results in heavy shear (AWSM vortex wake, U=8 m/s, $\alpha = 0.75$) for several radial stations against the theoretical momentum line. The reference wind speed for non-dimensionalizing Ct and a is now taken as the sector averaged wind speed

of the default Snel model for unsteady airfoil aerodynamics) in combination with the sector approach (ECNAero-BEM-BL-sector) results in unsteady loading characteristics matching AWSM very well for this case.

It is recommended to have a more detailed look into the definition of a representative streamtube (e.g. varying azimuthal extent and position leading/lagging, averaging procedure etc.) and run a variety of test cases (e.g. the cases were defined in Table 2 and the parametric shear investigation from Figure 3, containing a variation in the number of blades), also to assure this procedure does not cover up unintentionally other effects such as shed and trailed vorticity variation. Reference is made to a recent publication Madsen et al. (2019) that also addresses the issue of induction tracking by means of a new approach,

now solving the BEM equations on a polar grid. This approach seems to resolve the damping of local induced velocities by the dynamic inflow model by decoupling the individual blade momentum equations on a grid. In the current formulation this unwanted damping is counteracted by specifying a large number of subiterations per time step to ensure local convergence of the momentum equations, which is regarded as suboptimal.





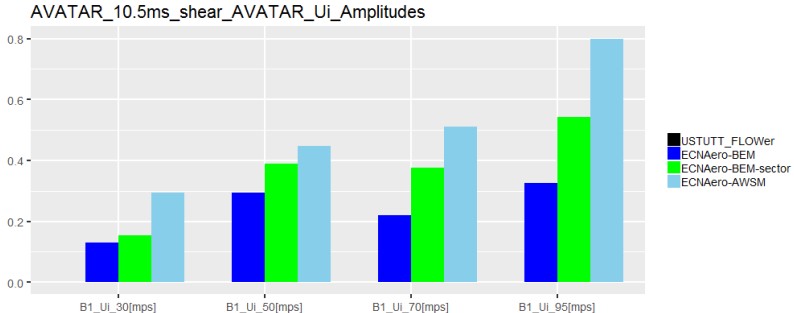

(a) Axial induced velocity amplitudes at 30%R, 50%R, 70%R and 95%R in shear (10.5 m/s)

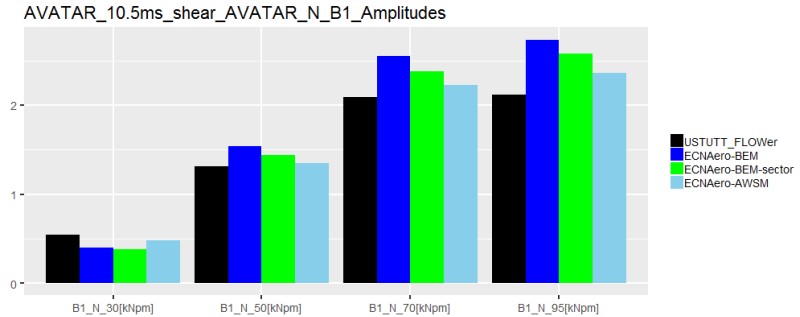

(b) Chordnormal force amplitudes at 30%R, 50%R, 70%R and 95%R in shear (10.5 m/s)

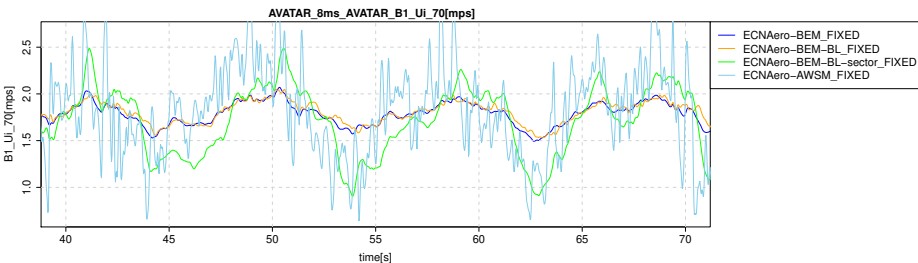

(c) Timetrace of axial induced velocity variation at 70%R in turbulent inflow (8 m/s)

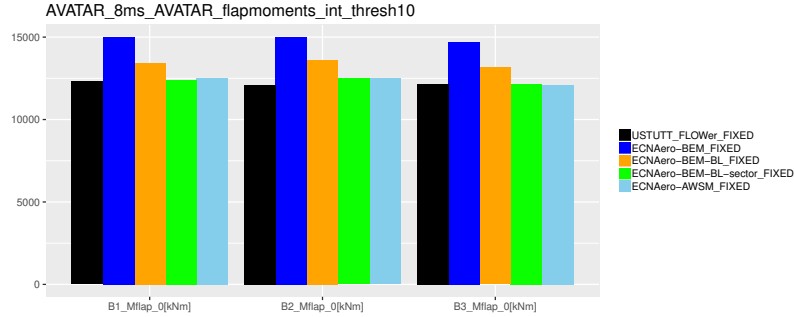

(d) Integrated staircase plot values (above a 10 count threshold) of flapwise blade root moment
in turbulent inflow (8 m/s) for all three blades.

**Figure 8.** Comparison of sector wind BEM implementation (sector) to conventional BEM with Snel and Beddoes Leishman (BL) modeling, vortex wake results (AWSM) and CFD (USTUTT_FLOWer) for selected AVATAR 10MW rotor simulations





# 3 Validation against field data

Over a decade of measurements on 2.5MW pitch to vane controlled research turbines is available from the EWTW test site Machielse, L.A.H. (2006). In an attempt to validate fatigue load predictions against field data, these measurements were subject of study.

## 3.1 Description of set-up

The EWTW farm Eecen et al. (2006) that is subject of investigation consisted of a row of five 2500 kW turbines with variable
speed-pitch regulated control. These turbines have a rotor diameter and hub height of 80 m and are placed at mutual distances of 3.8 rotor diameters (D). The farm is very well suited for investigation into effects at full scale because of its state of the art turbines and the comprehensive and reliable measurement infrastructure for turbine and meteorological data.

The farm was orientated from west to east (95-275°), see Figure 9. Turbine 6 has been instrumented with blade root strain gauges and hence is used for the loads analysis. The wind characteristics are measured with the meteorological tower at 2.5D
south-west of turbine 6. This mast measures wind speed and direction at three different heights including hub height. Also air pressure and temperature are measured at this height. More details can be found in the dedicated report Machielse, L.A.H. (2006). The analyzed measurements at EWTW have been obtained from the period September 2004 until January 2012.

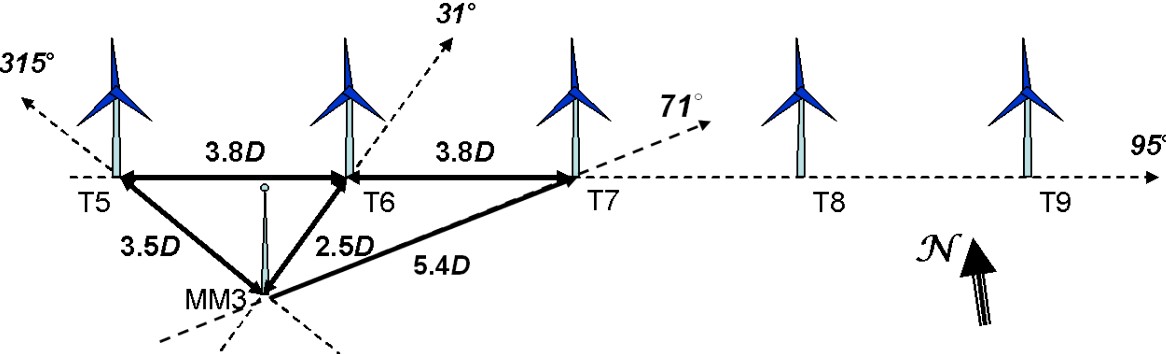

**Figure 9.** Main dimensions and directions in the EWTW farm. T5 to T9 are the turbine positions, MM3 indicates the measurement mast. Dimensions are expressed in rotor diameters D.

## 3.2 Data reduction

The SCADA and load signals of turbine 6 together with the meteorological data from mast 3 have been used for the analysis
in this report. 10-minute statistics have been retrieved from the data base. A wind direction criterium based on the undisturbed wind sector (between 110-140° and 200-250°) has been applied when retrieving the result from the database, resulting in about 100.000 samples. Further filtering out unwanted conditions (e.g. non-numeric values, start-up, stop or idling conditions) resulted in about 25.000 remaining 10-minute samples. The fatigue equivalent flapwise and edgewise moments of turbine 6





were acquired for for a slope of 10 (glass fibre). The rain-flow counting method was applied to the raw signal and the equivalent
loads have readily been determined in the database according to IEC 61400-13 iec (2001). Bin averaging is applied to the
resulting data sets both in wind speed and turbulence intensity. The standard error of the mean within each bin is calculated
using

$$S = \sigma/\sqrt{N} \quad , \tag{3}$$

with


$S$    []    standard error of bin average mean

$\sigma$    []    standard deviation of the bin data samples

$N$    [-]    number of samples per bin.

The resulting dataset from the filtering and binning has been visualized using contour plots as a function of turbulence intensity
and wind speed, e.g. for the fatigue equivalent flapwise blade root moment in Figure 10a.

### 3.3 Comparison to simulations

Using the bin averaged operational conditions from the field data analysis, simulations are performed for all wind speed bins
(5 to 12 m/s) focusing at the 10% turbulence intensity bin. A full aero-elastic model of the 2.5MW research turbine was built
using the PhatAero code as embedded in the FOCUS6 software, including mass, stiffness, control and aerodynamic details as
disclosed by the manufacturer. In order to create a representative value for the fatigue loads, six ten minute seeds were created
per wind speed bin using the Turbsim wind generator B.J. Jonkman and M.L. Buhl, Jr. (2006), making sure that the resulting
turbulence intensity matched the specification from the field data analysis. In view of the limited time, the amount of vortex
wake simulations (PhatAero-AWSM) was limited to only a few seeds. For each wind speed considered, a representative seed
was selected which matched the statistics and equivalent loads compared to the average over the six seeds for each wind speed
bin as good as possible. For these specific seeds, the rotational speed variations resulting from the BEM simulations were
recorded and fed to the AWSM simulations to have a consistent comparison between them. The settings were similar to the
settings as reported in section 2.1.4. The statistics and equivalent loading of all simulation results were obtained after skipping
the first 100 seconds, which is regarded as initialisation time, hence using the remaining 500 seconds. Similar to the binning
of the measured 10 minute statistics, the simulation results were averaged over the six available seeds for each wind speed
bin. In addition to that also the standard error was calculated in accordance with equation 3. The main comparison plot result
is given in Figure 10b. The results for the elected representative seeds are also given indicated by PhatAero-BEM-seeds and
PhatAero-AWSM-seeds.

The equivalent loading for the flapwise moments are over predicted around 15% by the BEM simulations (averaged over
all seeds and blades), where the AWSM vortex wake simulations are very close to the measurements (-1% averaged over all
seeds and blades). A similar conclusion can be drawn for the standard deviation. This trend is similar to the results obtained
from the comparison to CFD. Although the absolute difference between measurements and BEM simulations increases with
wind speed, the relative difference in terms of percentage remains largely constant over the wind speed range. It is noted that,



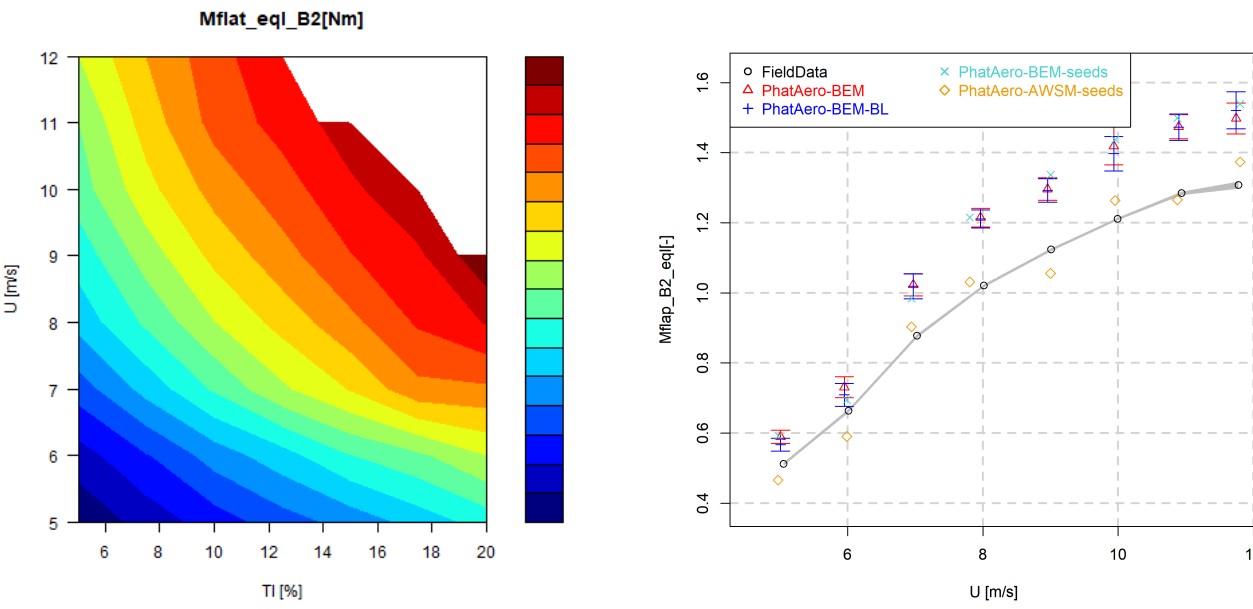

(a) Field data variation with wind speed U and turbulence intensity TI

(b) Comparison between field data and simulations at TI=10% as a function of wind speed (all values are non-dimensionalized using the average of the field data results over the wind speed bins)

**Figure 10.** Visualization of damage equivalent flapwise blade root moment

although not shown here, the averaged flapwise moments are slightly (<5%) underpredicted by the simulations, where they agree well between the different simulation settings.

Application of the Beddoes Leishman model (PhatAero-BEM-BL), which adds modeling of shed vorticity effects, reduces the difference with the measurements only slightly (around 1% decrease). This is not in agreement with the comparison to
CFD featuring the 10MW AVATAR rotor, which showed the modeling of shed vorticity to reduce the difference between BEM and high fidelity models significantly. It is unclear at this point what is causing the discrepancy between these observations.

Care should be taken drawing conclusions on the basis of these results, since it is felt that comparing aero-elastic simulations to field data is subject to many uncertainties (inflow, control, model data, compensating errors etc.) that cannot easily be verified. A great effort was made however to eradicate most of these, e.g. by running simulations for a large number of seeds and using
a large number of measurement samples. It is recommended to set-up a dedicated field test in an effort to further reduce the underlying uncertainties. Here one can think of using nacelle LiDAR to characterize the inflow conditions in more detail for synthetic wind field creation in combination with unsteady pressure sensors to measure sectional aerodynamic loading. In addition to that it is recommended to include more vortex wake simulations (similar to the number of BEM simulations) to better quantify the difference between these code types. More details about the comparison to field data can be found in the
dedicated report Boorsma (2019).





# 4 Impact on IEC design load calculations

Both vortex wake and BEM based models describe the blade aerodynamics on the basis of sectional properties of the airfoils, and both with options to account for dynamic stall effects and corrections for the effects of rotation. This means that the main difference between these model types is the description of the rotor wake aerodynamic effects, which is done in far more detail

by the vortex wake methods. Especially for operational conditions with asymmetric and non-uniform rotor loading the more detailed description of the wake influence may give a difference compared with BEM-based wake descriptions. An inventory is made from which conditions and IEC load cases Commission (2009) a difference is to be expected from vortex wake instead of BEM based models. Scoping analyses have been performed with both a BEM and vortex wake code for an entire fatigue load set to verify the differences.

## 4.1 Conditions and IEC load cases

Load case conditions for which vortex wake descriptions are expected to give more realistic predictions can be categorized as follows.

- Non-uniform inflow conditions

    The derivation of BEM theory assumes axi-symmetric (or uniform) inflow conditions with respect to the streamtube
considered. Sheared, turbulent or waked inflow conditions violate this assumption. As such vortex wake models are expected to provide a more accurate prediction for these conditions.

- Unsteady disk loading

    Many BEM based methods have a correction method for unsteady disk loading (e.g. pitch steps or coherent wind gusts) that is based on the influence of a cylindrical wake structure with constant wake diameter. For large disk loadings the
rotor wake expands, which gives additional non-linear contributions. So especially for high disk loading the dynamic inflow effects are predicted more accurate with vortex wake methods.

- Large yaw misalignment

    Most BEM based methods have a correction for the induced velocity distribution in oblique inflow. Most of these corrections are based on empirical fits of the asymmetric disk loading for a few rotors. A more accurate representation of
the effects of large yaw misalignment can be obtained with a vortex wake description.

- Asymmetric blade loads

    Asymmetry between the blade loads can be induced for example by a failed pitch actuator. This cannot be described with most of the BEM based approaches.

- Spanwise circulation variation

The effects of large gradients in spanwise circulation are not captured by BEM as there is no radial interaction. Examples are distributed control features (e.g. flaps) or angle of attack reduction towards the tip. For blades with a (nearly-) constant





chord and twist towards the blade tip, this effect can be described reasonably well in BEM based codes with the Prandtl tip loss factor. For an arbitrary tip shape (tapered, rounded and reverse twisted) the inflow angle reduction towards the tip can be described well with Vortex Wake methods.

– Radial induction

Rotors operating at a high disk loading feature wake expansion which is not captured in BEM based models. In cases where the blade has an orientation component perpendicular to the rotor plane (such as large cone angles), radial induction will start to influence blade loads.

Eventually it depends on the turbine under consideration (e.g. operating axial induction factor, blade shape), if the load cases 495 listed here give structural loads that are significant for the design. Based on the conditions described here the following load cases from IEC 61400-1 Commission (2009) are considered for evaluation with a vortex wake model.

  – DLC1.2 Normal Power production

In general the wake descriptions with vortex wake methods really make a difference if the induced velocities at the rotor are a significant fraction of the ambient wind velocity. This means for example that for a wind near or above the cut-out 500     conditions the wake effects have a very small contribution. Scoping analyses on the production load cases are reported in section 4.2.

  – DLC2.4 Operation with failed yaw or failed pitch that is not (yet) detected

These load cases contribute to the assessment of the fatigue loads. Both an undetected yaw failure and an undetected individual pitch failure give increased asymmetric loads over the rotor, for which a Vortex Wake description is more 505     detailed.

Prior to analysing some of the ultimate load cases with vortex wake programs, it is recommended to first analyse these load cases with a BEM based program to explore which of the load cases are design-driving. This holds especially for the cases with longer simulation time. Already it is envisaged that for the following load cases calculation with vortex wake models may be more realistic.

– DLC1.4 ECD

The direction change gives strong asymmetric loads that are also unsteady. For this condition a vortex wake model is more realistic, while the CPU effort is small for the short time span.

  – DLC1.5 EWS

Already for the shear of the NWP wind the BEM based calculations give larger blade root bending moment variations 515     than calculations with a vortex wake as was demonstrated in section 2.2.

  – DLC2.4

Operation with failed yaw or failed pitch that is not (yet) detected. These load cases contribute to the assessment of the





fatigue loads. Both an undetected yaw failure and an undetected individual pitch failure give increased asymmetric loads over the rotor, for which a vortex wake description is more detailed.

– DLC2.1 and DLC2.2

These cases involve blade failure or yaw failure and may be calculated with a vortex wake method because of the asymmetric rotor disk loading. This is especially the case if e.g. the actuator of one of the blades seizes or has a runaway that eventually triggers the controller to stop by pitching the other blades that do not have pitch failure.

– DLC3.3 Start with ECD

Here the direction change gives a strong asymmetric rotor disk loading while the start process itself can be considered as a transient. For this condition a vortex wake program is more realistic, while the time span of the simulation is relatively short.

– DLC6.2 50-year EWM after grid loss

The calculated loads for DLC6.2 tend to be design driving for some components. For DLC6.2 one usually has to assume
that the turbine is not yawing which means that the 50-year EWM may come from many directions and a serious set of calculations may be performed. Besides the fact that DLC6.2 may involve large yaw misalignment the influence of the wake is moderate because of the low rotor disk loading. Here one should realise that for a strong yaw misalignment, one or some of the blades are likely to get in stall which is hard to describe with both a BEM based program and with a vortex wake program.

– DLC6.3 1-year EWM with extreme yaw misalignment

Because of the strong yaw misalignment this load case may be calculated with a vortex wake model. Also here the limitations for both BEM and vortex wake descriptions apply if one or some of the blades get into stall.

One may expect that DLC1.3 ETM may be considered for calculation with a vortex wake model because DLC1.3 tends to give design driving loads and because the Extreme Turbulence Model may give highly non-uniform rotor disk loading that is quite
unsteady at the same time. Besides the large amount of CPU that is needed for the various 600s calculations one may argue that using a vortex wake model doesn't make much sense because eventually the turbulence level of the EWM has to be scaled such that the extreme blade root bending moments and the largest blade tip deformations match with the 50-year extrapolated values from DLC1.1. This means that if DLC1.3 is calculated with a vortex wake model instead of a BEM-based model, one may end up with a different scaling of the ETM. At least for the blade root bending moments and for the largest tip deformations the
use of a Vortex Wake model will not make a serious difference. Differences may be obtained in the other turbine components, if the recommended practice of the IEC is followed.

## 4.2 Fatigue load set

Scoping analyses have been performed with both BEM based programs and the vortex wake code AWSM for an entire fatigue load set featuring the 10MW AVATAR turbine. Design class IA was used which has a reference wind of 50 m/s, a Weibull





average wind of 10 m/s and a characteristic turbulence level of 16%. The wind is modelled with a power law for the vertical shear with exponent 0.2, although for offshore wind turbines the IEC recommendations prescribe a vertical shear exponent of 0.14. The inclination of the ambient wind is set to zero. The wind velocities for which the turbine is in operation range from 4 m/s through 25 m/s while for each wind (with 1 m/s intervals) three calculations are performed with different wind stochastics. For these three calculations the yaw misalignment has values of -8°, +8° and 0°. The turbulence applies to the frequency

spectrum of Kaimal and for each wind velocity another random seed was used. In comparison to the rigid rotor calculations which were compared to CFD, the aero-elastic turbine in Phatas was modelled including all flexibilities (e.g. tower and blades) and active controller as defined in the EU AVATAR project. The time increment was set to 0.05s. Simulations were performed with Phatas, Phatas-BSV, PhatAero-BEM and PhatAero-AWSM.

Although the resulting load characteristics were also compared for the tower and nacelle, the results of the flapwise blade root moments are given in Figure 11. This figure shows the largest reductions of the vortex wake model in sub-rated conditions

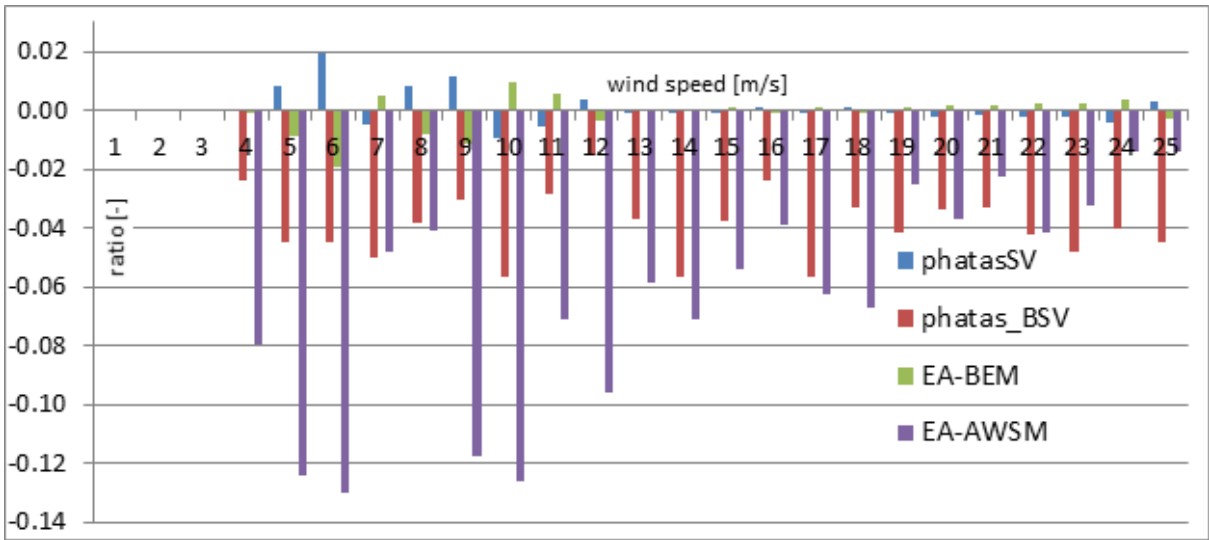

**Figure 11.** Ratio of blade root flap fatigue loading relative to the average of phatasSV and PhatAero(EA)-BEM as a function of wind speed


featuring higher axial induction factors, in agreement with the observations from section 2 for the comparison to CFD. For wind velocities above rated the fatigue predictions with blade shed vorticity model (Phatas-BSV) are close to the AWSM vortex wake simulations.

Although only a set of normal production load cases is calculated with AWSM it is expected that the reduction in overall

fatigue damage by using the program AWSM may be up to 5% for the AVATAR rotor. This is a consequence of the relative large contribution of the higher wind speeds to the overall fatigue. The blade shed vorticity algorithm gives an overall fatigue load reduction of about 2% compared with the BEM based programs without this blade shed vorticity contribution. It is noted that the given percentages are obtained for the AVATAR turbine featuring a low induction rotor and may vary depending on the





design operating axial induction. For more details please consult the dedicated report about the IEC load set survey Lindenburg
570 (2019).

## 5 Conclusions

Making a meaningful comparison between CFD (numerical wind tunnel) and lifting line codes has appeared to be quite a
challenge in terms of inflow alignment. However a promising engineering approach was devised which allowed successful
comparisons in the time domain between these code types. A test matrix was defined covering representative operational
and inflow conditions, bearing the CPU requirements in mind. The fatigue load reduction from BEM to vortex type codes
as observed in the EU AVATAR project has been confirmed by these dedicated CFD simulations. Partly this is explained by
the shed vorticity effect which is implicitly included in the vortex wake type codes, but poor tracking of wind variations by
induction for BEM remains an issue. It is recommended to further study this aspect to further reduce uncertainties in BEM
modeling. In addition to that very similar results were obtained between several vortex type codes originating from different
institutions. A variety of load cases has shed more light on this subject, showing a correlation of the observations with axial
induction factor.

In addition to the comparison against CFD simulations, a validation was made against measured fatigue loads of a real
turbine at the EWTW test site. Over 7 years of measurements were analysed to obtain relevant statistics over 100.000 ten
minute samples, of which about 25.000 remained after filtering out unwanted conditions. The data was bin averaged with
respect to turbulence intensity and wind speed, after which dedicated simulations for each wind speed bin were ran at 10%
turbulence intensity. The resulting load comparison shows BEM to over predict the fatigue equivalent flapwise blade root
moments, where a vortex wake model comes closer to the measurements. However care should be taken drawing conclusions,
since it is felt that comparing aero-elastic simulations to the field data set is subject to many uncertainties (inflow, control, model
data, compensating errors etc.) that cannot easily be verified. A great effort was made however to eradicate most of these, e.g.
by running simulations for a large number of seeds and using a large number of measurement samples. It is recommended to
set-up a dedicated field test in an effort to further reduce these uncertainties, allowing a better validation. Here one can think of
using nacelle LiDAR to characterize the inflow conditions in more detail for synthetic wind field creation in combination with
pressure sensors to measure sectional aerodynamic loading.

An inventory is made from which conditions and IEC load cases a difference is to be expected from vortex wake instead of
BEM based models. Based on past experience it is anticipated that differences are to be expected in non-uniform and yawed
inflow conditions especially when operating in high thrust coefficients. Scoping analyses have been performed with both a
BEM and vortex wake code for an entire fatigue load set to verify the differences. Although only a set of normal production
load cases is calculated with a vortex wake code it is expected that the reduction in overall fatigue damage by using a vortex
wake program may be up to 5% for the relatively low induction AVATAR rotor, of which about half was attributed to shed
vorticity effects. A more extensive exploration of design load calculations and the added value of vortex wake calculations still
is to be performed, not only focusing on fatigue loads but also on extreme loads and power production.



Concluding a very successful validation of lifting line codes against a 'numerical' wind tunnel and field data has been performed. A validation study similar to the cases studied in this project in a physical wind tunnel is recommended as 'proof of the pudding'.

*Author contributions.* K. Boorsma assembled the simulation results and ran the ECN.TNO simulations. F. Wenz performed all CFD simulations. C. Lindenburg ran the Phatas simulations and performed the survey over the production load set. M. Aman and M. Kloosterman contributed with the Bladed 4.8 simulations.

*Competing interests.* The authors declare that they have no conflict of interest.

*Acknowledgements.* This contribution has been sponsored within the framework of TKI Wind op Zee.

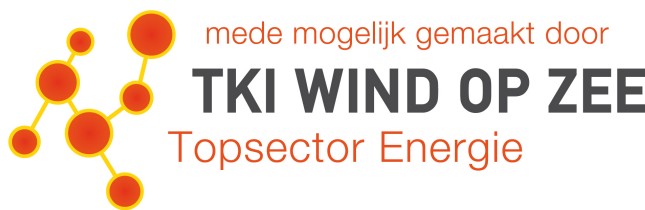




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
