# Peer review of "Validation and accommodation of vortex wake codes for wind turbine design load calculations"

_Wind Energy Science, 2020_

## Referee Comment (RC1) · Anonymous Referee #1 · 28 Jan 2020

This paper presents: *A comparison between CFD, vortex-wake methods, and BEM results, including the influence of shear and turbulence, with suggestions for how to improve lower-fidelity solutions based on the insights gleaned from higher-fidelity solutions *A validation of the BEM and vortex-wake methods against measured field data from a full-scale wind turbine. *Recommendations on when to apply vortex-wake methods based on operational condition and IEC design load case.

Overall, the paper is well written, the results appear to be scientifically sound, and the results are informative. A few corrections and clarifications are warranted to approve the final publication. Please find specific comments and technical corrections below:

Specific Comments (Section/ Line / Comment): - / - / The code-to-code results of section 2 seem very valuable for other BEM and vortex methods to compare to. Can these results be shared publicly, e.g., through a website? 2.1 / - / A couple comments. The general capability of each model is less important than the specific models employed in the study; suggest focusing only on the specific models employed. Also, for the BEM and vortex methods, it would be useful to add a table comparing/contrasting which models are employed, as well as the time/space discretizations used. The various subsections are a bit mixed now. 2.3 / 209 / The text says shear is neglected from case Nr 1. Is that the same for all cases? 2.3 / - / It would be useful to add a table comparing the computational expense, including real time and number of cores. 2.3 / 216 / Were the rotor speed and pitch derived from one simulation and applied to all simulation model, or just applied to the CFD? It would be preferred if they were applied to all simulations. 2.4 / 368 / It would be useful to add a figure to clarify this approach of sector averaging. 3.2 / 413 / What is meant by "standard error the mean" when sigma contains the standard deviation? 3.2 / 422 / Where the data provided by the manufacturer (especially aero) calibrated based on the field measurements? 3.2 / 425 / Was just the TI matched, or also the mean profile (shear), turbulence spectra, spatial coherence, Reynolds stresses? 4.1 / - / It would be also useful to mention that geometric details such as curvature, sweep, and/or deflection are also better captured by vortex methods over BEM. 4.1 / - / This discussion is useful, but perhaps a bit out of place considering that no results are presented showing the differences between BEM and vortex methods for these load cases. Perhaps save this for a future publication where results will be presented? 4.1 / 528 / DLC 6.X consider the turbine parked or idling, where the wake is expected to be minimal. Wouldn't a geometric AoA (without induction) suffice? 4.2 / 560 / The reviewer is unclear what is presented here. What is meant by "relative to the average..."?

Technical Corrections (Section / Line / Comment): 1/ 17 / The first two sentences are a bit odd for an international journal. Suggest making more generic or discussing international growth. 2.1.4 / 153 / Add a space between "model" and "Snel". 2.3.2 /

249 / Change "serie" to "series". 2.3.3/ - / The data of Table 3 would be interpreted better as a bar chart. 2.4 / 349 / The "W" in atan2 should be changed to "U". 2.4 / 353 / The reviewer's understanding is that these simulations considered a rigid structure, so, there is no torsion deformation. 3.2 / 409 / Clarify that "m=10". 3.2 / 424 / Change "Turbsim" to "TurbSim". 3.2 / - / Are the units on the legend purposely missing, e.g., to normalize the data?
* * *

---

## Referee Comment (RC2) · Emmanuel Branlard (Referee) · 12 Mar 2020

The paper presents a comparison of different classes of codes, evaluate differences in fatigue loads, suggests potential reasons for the differences observed, and present recommendations to use vortex wake models for some DLC loads cases. The study is thorough and convincing.

Here are my general comments:

- Shed vorticity is suggested as an explanation for the differences. Time constants of the dynamic stall and dynamic wake models are also likely to be a source of the "axial"

[Figure]

induction filtering observed. I would actually think, the dynamic wake model might be the main source of error, since as you mentioned it is tuned to "cylindrical" wakes. You can investigate the effect of the time constants of these models on the axial induction to see if you can reach amplitudes similar to the ones observed for the vortex wake codes.

- I would recommend to make a distinction between "vortex wake models" and "vortex methods" early on in the paper, or at least mention that what is meant by "vortex wake model" in the paper is low-order vortex filament methods. Vortex methods, in general, falls in the realm of CFD, and can be of the same order of accuracy as "traditional velocity-pressure CFD".

- I'm not sure I see the need to use the term "numerical wind tunnel", verification against "CFD simulations" seem more appropriate. It could be erroneously assumed that a "numerical wind tunnel" models walls, turbulence grid, etc.

- The presentation of the different codes are relevant and provide the appropriate details, yet I would recommend removing the mention of features that are not used in this study to avoid confusion. A table providing the differences in features (e.g. dynamic stall, high-thrust corrections, discretization, regularization parameters, viscous models, etc.) used by each code-class (BEM/Vortex) could be relevant as an introduction/conclusion of the second section.

- Some of the figures can be improved for readability, and consistency (underscore are sometimes present, and units are not always present on the y-axis, but mentioned on the x axis). Since the study present differences between BEM and Vortex methods, it might be worth distinguishing between these two classes of codes in the figures. Even if a consistent color scheme has been used throughout the paper, having a way to clearly see which class of code is used would help the reader (with dashed lines or markers maybe). In general, I would think the explanations in the figure captions could be slightly extended.

- Last, I believe any attempt to shorten the length of the paper would be beneficial.

I'll be happy to review a revised version of this manuscript.

Good luck for the remaining work,

Emmanuel

Here are my specific comments:

l78-84: Reporting the distances in diameter would be valuable.

l108: "dimensional form instead of nd factors": This is probably not as relevant as the form of the equations itself, I don't think it's needed to mention this.

l121: The term Phatas is used in the text whereas PhatasSV is used in the figure. Using one of the two terms throughout the document would avoid confusion.

l129-134: Since different features are listed here, it is not clear which ones will be used. I'd recommend to only mention the ones that are used in this study and provide a link to a documentation for more models.

l140: Can you discuss the regularization/"viscous-core" model used for the bound circulation and for the wake: which model is used, how the regularization parameter is defined, does it include a "diffusion" model?

l145: Can you mention the typical number of filaments in the wake?

l159: Maybe "blade averaged induction" need to be reformulated since it seems to apply to the wake.

l250: It might not be obvious what "fixed" stands for (it's mentioned on the next line).

l251-255: From figure 4a, it appears that the axial induction from vortex wake codes are significantly larger than the ones from the BEM codes. Can you expand your justification for the fatigue loads to be lower in this case? I would have expected the angle of attack fluctuations to be larger and the loads higher as well, but I might have

[Figure]

missed something.

Figure 1: Could you provide the tangential force and C_T as well for all figures? The normal force is usually well captured. Depending on how these other component vary with the operating condition, it might not be necessary to show the radial distribution for the 4 wind speeds, 2 seem to be sufficient (at least looking at Fn).

Figure 1: Even if the axial induction is not present in CFD, it will be valuable to show this variable for the lifting-line codes since this is the core variable here (the rest of the angle of attack is purely defined by the free stream and the rotational speed).

Figure 2: It would be relevant to show differences in mean as well as differences in amplitude.

Figure 4, 5: the caption should preferably mention what is meant by FIXED and BSV, BL_sector

l299: "structural" might be confusing here.

l346: the tip-loss factor has also been ignored

l560: Could you mention some of the results from the tower loads?

---

## Author Comment (AC1) · 27 Mar 2020

Many thanks to the reviewers for their valuable suggestions!

**Review 1**

Specific Comments (Section/ Line / Comment):

- / - / The code-to-code results of section 2 seem very valuable for other BEM and vortex methods to compare to. Can these results be shared publicly, e.g., through a website?

Results are available by emailing the corresponding author. This info has been added in the data availability section.

2.1 / - / A couple comments. The general capability of each model is less important than the specific models employed in the study; suggest focusing only on the specific models employed. Also, for the BEM and vortex methods, it would be useful to add a table comparing/contrasting which models are employed, as well as the time/space discretizations used. The various subsections are a bit mixed now.

A table with a high level overview is added.

2.3 / 209 / The text says shear is neglected from case Nr 1. Is that the same for all cases?

Yes this is the case and is now more explicitly stated in the paper.

2.3 / - / It would be useful to add a table comparing the computational expense, including real time and number of cores.

Yes this would be very useful but the info is not really available

2.3 / 216 / Were the rotor speed and pitch derived from one simulation and applied

to all simulation model, or just applied to the CFD? It would be preferred if they were

applied to all simulations.

From 1 simulation and applied to all models. This is clarified better in the text now.

2.4 / 368 / It would be useful to add a figure to clarify this approach of sector averaging.

Illustration is added to figure 7

3.2 / 413 / What is meant by "standard error the mean" when sigma contains the standard deviation?

The standard error of the mean within each bin is a measure for repeatability of the mean values collected within a bin. It is common practice in measurements to define it as equation 3 by dividing the standard deviation over the number of samples.

3.2 / 422 / Where the data provided by the manufacturer (especially aero) calibrated based on the field measurements?

No not as far as I am aware of. Reworded this sentence to clarify that blade data was delivered separately by the blade manufacturer.

3.2 / 425 / Was just the TI matched, or also the mean profile (shear), turbulence spectra,

spatial coherence, Reynolds stresses?

Only TI was matched. Sentence was added to clarify better.

4.1 / - / It would be also useful to mention thatgeometric details such as curvature, sweep, and/or deflection are also better captured by vortex methods over BEM.

Section 4.1 has been shortened

4.1 / - / This discussion is useful, but perhaps a bit out of place considering that no results are presented showing the differences between BEM and vortex methods for these load cases. Perhaps save this for a future publication where results will be presented?

Section 4.1 has been shortened

4.1 / 528 / DLC 6.X consider the turbine parked or idling, where the wake is expected to be minimal. Wouldn't a geometric AoA (without induction) suffice?

Section 4.1 has been rewritten

4.2 / 560 / The reviewer is unclear what is presented here. What is meant by "relative to the average..."?

A ratio is always relative to a certain number which in this case is the average over the two BEM simulations. Made a rewording attempt to better clarify this.

Technical Corrections (Section / Line / Comment):

1/ 17 / The first two sentences are a bit odd for an international journal. Suggest making more generic or discussing international growth.

Sentences replaced

2.1.4 / 153 / Add a space between "model" and "Snel".

ok

2.3.2 / -249 / Change "serie" to "series".

ok

2.3.3/ - / The data of Table 3 would be interpreted better as a bar chart.

Agree so modified

2.4 / 349 / The "W" in atan2 should be changed to "U".

Good spot!

2.4 / 353 / The reviewer's understanding is that these simulations considered a rigid structure,

so, there is no torsion deformation.

Indeed the cases featured in section 2.4 are rigid. The equations given however are intended for any general case.

3.2 / 409 / Clarify that "m=10".

Text modified

3.2 / 424 / Change "Turbsim" to "TurbSim".

ok

3.2 / - / Are the units on the legend purposely missing, e.g., to normalize the data?

Yes to comply with confidentiality (now indicated in the text)

**Review 2**

The paper presents a comparison of different classes of codes, evaluate differences in fatigue loads, suggests potential reasons for the differences observed, and present recommendations to use vortex wake models for some DLC loads cases. The study is thorough and convincing.

Here are my general comments:

- Shed vorticity is suggested as an explanation for the differences. Time constants of the dynamic stall and dynamic wake models are also likely to be a source of the "axial" induction filtering observed. I would actually think, the dynamic wake model might be the main source of error, since as you mentioned it is tuned to "cylindrical" wakes. You can investigate the effect of the time constants of these models on the axial induction to see if you can reach amplitudes similar to the ones observed for the vortex wake codes.

Indeed the dynamic wake model is also playing a role here, which was explained in section 2.2 in sheared inflow. The explanation has been expanded to section 2.3.2 as well to make the point more prominent.

- I would recommend to make a distinction between "vortex wake models" and "vortex methods" early on in the paper, or at least mention that what is meant by "vortex wake model" in the paper is low-order vortex filament methods. Vortex methods, in general, falls in the realm of CFD, and can be of the same order of accuracy as "traditional velocity-pressure CFD".

Clarification added in introduction

- I'm not sure I see the need to use the term "numerical wind tunnel", verification against "CFD simulations" seem more appropriate. It could be erroneously assumed that a "numerical wind tunnel" models walls, turbulence grid, etc.

Changed to CFD to prevent misinterpretation

- The presentation of the different codes are relevant and provide the appropriate details, yet I would recommend removing the mention of features that are not used in this study to avoid confusion. A table providing the differences in features (e.g. dynamic stall, high-thrust corrections, discretization, regularization parameters, viscous models, etc.) used by each code-class (BEM/Vortex) could be relevant as an introduction/conclusion of the second section.

A table with a high level overview is added

- Some of the figures can be improved for readability, and consistency (underscore are sometimes present, and units are not always present on the y-axis, but mentioned on the x axis). Since the study present differences between BEM and Vortex methods, it might be worth distinguishing between these two classes of codes in the figures. Even if a consistent color scheme has been used throughout the paper, having a way to clearly see which class of code is used would help the reader

(with dashed lines or markers maybe). In general, I would think the explanations in the figure captions could be slightly extended.

Explanations in captions have been extended

- Last, I believe any attempt to shorten the length of the paper would be beneficial. I'll be happy to review a revised version of this manuscript.

Paper has been shortened slightly by restructuring section 4.1

Good luck for the remaining work,

Emmanuel

Here are my specific comments:

l78-84: Reporting the distances in diameter would be valuable.

Info added in table 1

l108: "dimensional form instead of nd factors": This is probably not as relevant as the

form of the equations itself, I don't think it's needed to mention this.

l121: The term Phatas is used in the text whereas PhatasSV is used in the figure.

Using one of the two terms throughout the document would avoid confusion.

Made consistent throughout the paper

l129-134: Since different features are listed here, it is not clear which ones will be used.

I'd recommend to only mention the ones that are used in this study and provide a link

to a documentation for more models.

All the ones mentioned here are used (dynamic stall and rotational effects excepted)

l140: Can you discuss the regularization/"viscous-core" model used for the bound circulation and for the wake: which model is used, how the regularization parameter is defined, does it include a "diffusion" model?

These parameters are typically important for wake studies (or wake interaction with a downstream rotor) but not really relevant for the loads studied in this project.

l145: Can you mention the typical number of filaments in the wake?

Info is added for the ECNAero-AWSM simulation. Because AWSM is also used when coupled to PhatAero, the order of these two code description sections has been swapped.

l159: Maybe "blade averaged induction" need to be reformulated since it seems to apply to the wake.

Reformulated

l250: It might not be obvious what "fixed" stands for (it's mentioned on the next line).

Order changed

l251-255: From figure 4a, it appears that the axial induction from vortex wake codes are significantly larger than the ones from the BEM codes. Can you expand your justification for the fatigue loads to be lower in this case? I would have expected the angle of attack fluctuations to be larger and the loads higher as well, but I might have missed something.

The variation in axial induced velocity is larger for vortex wake codes as it follows the inflow variations better. The resulting velocity triangle for a section will show the angle of attack to be more constant because the increase in wind is (partially) compensated by the increase in axial induction.

Figure 1: Could you provide the tangential force and C_T as well for all figures? The normal force is usually well captured. Depending on how these other component vary with the operating condition, it might not be necessary to show the radial distribution for the 4 wind speeds, 2 seem to be sufficient (at least looking at Fn).

Tangential force added for 2 velocities

Figure 1: Even if the axial induction is not present in CFD, it will be valuable to show this variable for the lifting-line codes since this is the core variable here (the rest of the angle of attack is purely defined by the free stream and the rotational speed).

As there is a good agreement in terms of forces as well as axial induction this variable is not shown here.

Figure 2: It would be relevant to show differences in mean as well as differences in amplitude.

The uniform inflow case was to establish agreement in terms of mean level, the sheared case focused on amplitudes. To reduce the nr of pages this information is left out here, but is available from the dedicated Vortexloads report.

Figure 4, 5: the caption should preferably mention what is meant by FIXED and BSV, BL_sector

Info is added

l299: "structural" might be confusing here.

Removed structural

l346: the tip-loss factor has also been ignored

Added in the sentence

l560: Could you mention some of the results from the tower loads?

Comment added

**Reviewer 3**

Dear authors,

this is a very interesting article on the validation of different aerodynamic codes for load calculations. Especially the comparison with field data is very impressive, and it is great how the reasons for differences between results from different codes are investigated in detail. However, there are some points where I don't quite understand what exactly is compared and how some of

the models work. I am also suggesting a couple of references to Section 4 below, it's up to you if you want to include them.

Thank you for the good work! Please find my comments below.

**General comments**

* As a general comment there are many places in the article where \citep might be a better option than \cite.

Thanks for this great advice!

* An overview table of what kinds of corrections / models are used in the different BEM codes could be valuable.

Table with high level overview of codes was added

* It could be great if some of the data from the article could be shared. It would be very interesting for others to test their models in some of the cases and compare to your results as well as the CFD!

Results are available by emailing the corresponding author. This info has been added in the data availability section.

**Specific comments**

* Section 2.1.1 FLOWer: You write on line 95 that the airfoil data comes from CFD simulations. It's definitely a good idea to use 2D data on the outboard part for the vortex wake simulations. A few questions:

      * Do you also use 2D data outside of 70% for the BEM simulations?

Yes

      * Is it correctly understood that 3D polars have only been computed for the comparisons on the AVATAR turbine? So the comparison with measurements in Section 3 uses 'conventional' 2D airfoil data?

Yes

      * Can you indicate at which AOA range you computed the CFD polars? And what did you do outside of that AOA range?

The 3D CFD polar data was extracted using the azimuthal averaged approach by varying the inflow speed while maintaining the rotational speed constant. This results in different polar range depending on the location of the airfoil section. In the inboard area, for example, the maximum angle can reach as large as 40 deg, while in the outer part it is smaller than that. As the main purpose for this dedicated approach is to correct the 3D effects in the root area of the blade, the extracted AoA range is sufficient. To cover for even larger AoA range, the original polar dataset of the AVATAR turbine was applied.

      * As you write these polars include some 3D effects. Have you seen issues when using these polars up to 16 m/s with fairly high turbulence intensity? I would think these polars depend on the load distribution and at higher wind speeds the maximum loading and also the tip vortex will shift further inboard. Also the root vortex will become more important, so there may be an argument for

using 2D data closer to the root as well (If you have a root loss model in a BEM or when computing with AWSM).

Engineering model calculations employing the extracted 3D CFD polar perform very well in comparison to the fully resolved CFD data even for U = 25 m/s. This was demonstrated in a previous report (Page 18, Figures 17 and 18) [1], not only in terms of the integral loads but also in terms of the sectional loads. Root effects indeed strictly mean that 2D data needs to be used here as well but the project focus is not in this area.

 [1] Bangga, Galih. "Comparison of blade element method and CFD simulations of a 10 MW wind turbine." Fluids 3, no. 4 (2018): 73.

References have been added

       * Could the 3D CFD polars be made available?

Yes by emailing corresponding author who will redirect you to the responsible at Stuttgart Uni

* Section 2.1.2 Bladed 4.8

       * Can you explain how the induced velocities are computed in Bladed? Is it an azimuth averaged approach? At least Figure 2 b) suggests that.

Section 2.2 explains this in more detail: "Application of the dynamic inflow model to the local element induced velocity (as implemented in the Bladed4.8-BEM results following the TUDK model as described in (Snel and Schepers, 1994)) appears to dampen out induced velocity variations in non-uniform inflow conditions"

* Section 2.1.3 Phatas

       * I don't think I understand the purpose of the blade shed vorticity model in comparison with the 2D attached flow part of the Beddoes-Leishman dynamic stall model. It seems to me that

              * Both model only the effect of the blade sections on themselves

              * Both model the effect of the shed vorticity only (due to the time derivative of the circulation) at each section

              * The Beddoes-Leishman model is probably easier to implement and faster to compute.

       * Maybe I am missing the point but what is the advantage of the BSV model over the Beddoes-leishman model? Is it that the BSV model takes the finite blade length into account towards the tip? I am not sure I understand that from the reference. If it is about the finite blade length I would expect it to predict a smaller effect than the BL model (which is 2D), but in Figure 5 the effect of the BSV model seems to be a bit larger instead.

Since they are 2 models simulating the same thing (one acts on force coefficients and the other on induced velocity) it is interesting to compare.

       * I assume that the dynamic stall model is without the second order terms that cause force variations for constant inflow and AOA in stalled conditions based on the strouhal number, is that correct?

Yes (clarification added)

* Section 2.2 Constant uniform and sheared inflow

    * Figure 2 a) Why are no flower results shown here?

They are there in black but the legend did not indicate it (fixed now).

    * line 193, 'calculations were done for various conditions'. A short summary of these conditions would be helpful to understand where the results in Figure 3 come from.

Wind speed range was added

    * line 200, The trend looks quadratic to me.

Observation modified

* Section 2.3.1 Comparison methodology

    * It sounds very promising how you made sure that the turbulence in CFD and lifting line codes matches. Is it possible to show a plot of the wind speed comparisons mentioned in line 240 to see how good the agreement is?

To reduce the number of pages this plot is shown in the dedicated Vortexloads report which is referenced

* Section 2.3.3 Effect of load case variations

    * Table 3: Why not show the EQLs in the same way as the INT and compare all three codes to CFD?

As mentioned: 'Although comparison of equivalent loads between CFD and lifting line codes is hindered by small differences in inflow conditions, a comparison between lifting line codes (BEM and vortex) in terms of fatigue loading is deemed useful in the last two columns of Table 3'

* Section 2.4 Improved induction tracking

    * line 299: Maybe 'fundamental difference' instead of 'structural difference'? I'm not quite sure.

Modified

    * line 303: I would also expect that there will not be a big shed vorticity effect in vertical shear, because the time constants of the shed vorticity are in the order of chord length/(2 relative velocity). This is much faster than the 1P variations due to shear for most of the blade.

    * Equation (1): I would prefer $C\_t$ and $U\_i$ instead of $Ct$ and $Ui$

    * Equation (2): Maybe you could name these wind speeds 'U' differently to make it clear that you use a wind speed for example 'U_m' for the momentum equations on the left side and 'U_f' for the force on the ritght side. These could then also be used in the text making it easier to understand how exactly the sector approach works.

This good idea is adapted

    * Figure 6:

        * Are you using local values for Ct and a? I think it is very important to be as specific as possible here.

Extra line is added to clarify how Ct and a are calculated

        * Can you add points for BEM close to the tip (as you have for AWSM on the right side)? I would expect that the tip correction would make it not fall on the line for the momentum equations, and that would be a good thing.

Because the tip correction is used in the definition of a, the tip values are still on the momentum line and not shown here.

    * line 367: Is it correctly understood that you use the sector wind speed 1) on the left hand side of Equation (2) and 2) in the computation of the induced velocity in Equation (1)? I think this should be written clearly in the text (possibly by also using for example 'U_m' in Equation (1) as suggested above for Equation (2).

Suggestion adopted

    * line 386: I am not sure I understand what you mean by subiterations to ensure convergence of the momentum equations to avoid a lag due to the dynamic inflow. I think what the dynamic inflow model is supposed to do is to introduce time lags so that there is no equilibrium between the local forces and the induced wind at each time step. The momentum equations will only be converged in steady state. In our grid BEM implementation each point in a pure shear case is constantly in a steady state, that is why the dynamic inflow model is not contributing. Would your subiterations change the behaviour of the model for instance in the case of a pitch step in uniform inflow?

I think the initial text was unclear in the sense that I donot mean convergence between forces and induced wind only, but convergence between forces, induced wind and the dynamic inflow term. In our case this is not guaranteed because we have the annulus averaged induction (Ui_b1+Ui_b2+Ui_b3)/3 as convergence criterium. The text is rephrased to make this more clear.

* Section 3.3 Comparison to simulations

    * line 445: Are you doing something in your implementation to 'deactivate' the shed vorticity part in stalled flow? Otherwise some unphysical lags of the effective AOA behind the geometric AOA in stall might cause some increased load spikes. We have tried to deal with that in 'Pirrung, G., & Gaunaa, M. (2018). Dynamic stall model modifications to improve the modeling of vertical axis wind turbines. DTU Wind Energy E, No. 171'. Not sure if this is responsible for the overall smaller than expected load reduction though, just an idea.

No specific implementation is done for shed vorticity in stalled flow. Thanks for this suggestion.

* Section 4 Impact on IEC design load calculations

    * I think in general it could be nice to add some references here. I completely agree that vortex codes could give improved results in all the conditions you list, but there has been quite some effort in improving BEM modeling so some of the shortcomings of BEM are probably not as severe as they used to be.

Section 4.1 has been rewritten

    * Line 472

        * Suggested references: Wei Yu's work on dynamic inflow, our BEM paper, the Mexnext III final report, your torque paper from 2016

* Line 480 on large yaw misalignment

* Maybe a reference to the final report of mexnext 3 that includes some yaw comparisons would make sense

* Line 485

* High wind speeds are also a prime example of spanwise circulation variation, where the loading on the disc is far from uniform and Prandtl tip loss doesn't apply.

* I think a reference to our near wake model work that is directly adressing the missing spanwise coupling in BEM codes might be relevant here

* Line 490

* I think a reference to the radial induction model in our BEM paper or to Aagaard Madsen, H., Bak, C., Døssing, M., Mikkelsen, R. F., & Øye, S. (2010). Validation and modification of the Blade Element Momentum theory based on comparisons with actuator disc simulations. Wind Energy, 13(4), 373-389. could be relevant here

* line 500

* Are you sure that the wake effects are small? I agree that the complete induction is small but I think the effects at the individual blades due to tip/root loss can be substantial.

* line 532

* Same as just above: The disk loading is low but there may still be strong vorticity trailed from the blades that locally changes AOA quite a bit. What is important is that it can change whether parts of the blade are in stall or attached flow, which changes the aerodynamic damping of for example edgewise vibrations in stand still. You might name specifically that vortex induced vibrations may occur in 6.2, which can't be predicted by BEM or vortex codes.

* line 558 is the phataero-BEM including beddoes-leishman dynamic stall?

No

---

## Author Response (AR2)

15/05/2020

Many thanks to the Associate editor for taking the time to have a look at this paper. Please find below the reply of the authors in red. A pdf which shows the changes with respect to the previous version is uploaded as well.

Section 1 Introduction

Lines 16 – 21: Motivation could use a bit of work. It is not bad but it could be better. Offshore is not the only place where increasing rotor size and slender flexible blades present design challenges. The square cube law is introduced without context – most readers should be familiar but not all. Transition to introducing aerodynamic model types is abrupt

Lines 26-29: again, a little weak. The author assumes a lot of knowledge of the readers about existing turbine design practice.

The first part of the intro (line 16-29)  has been rewritten to take into account the suggestions.

Line 30 – can you explain in one sentence WHY vortex codes are more expensive?

The sentence has been modified to clarify this point.

Section 2

The AVATAR 10 MW turbine is introduced without any details. Given the motivation around large rotors with slender flexible blades, it would be nice to know something about the design features of that machine – how stiff are the blades? How large is the chord relative to typical turbines? Any novel features like aeroelastic tailoring or other things that would be good to know? Basic details of rotor diameter, specific power, etc.?

Several relevant details of the turbine are added.